# Summertime response of ozone and fine particulate matter to mixing layer meteorology over the North China Plain

Jiaqi Wang[1], Jian Gao[1], Fei Che[1], Xin Yang[1], Yuanqin Yang[2], Lei Liu[2], Yan Xiang[3]

[1]State Key Laboratory of Environmental Criteria and Risk Assessment, Chinese Research Academy of Environmental Sciences, Beijing 100012, China

[2]State Key Laboratory of Severe Weather and Key Laboratory for Atmospheric Chemistry of CMA, Chinese Academy of Meteorological Sciences, Beijing 100081, China

[3]Institutes of Physical Science and Information Technology, Anhui University, Hefei, China

*Correspondence to:* Jian Gao (gaojian@craes.org.cn)

**Abstract.** Measurements of surface ozone ($O_3$), $PM_{2.5}$ and its major secondary components ($SO_4^{2-}$, $NO_3^-$, $NH_4^+$, and OC), mixing layer height (MLH) and other meteorological parameters were made in the North China Plain (NCP) during warm seasons (June–July) in 2021. The observation results showed that the summertime regional MDA8 $O_3$ initially increased and reached the maximum (195.88 $\mu g\ m^{-3}$) when MLH at around 900–1800 m, then turned to decrease with further evolution of MLH. Interestingly, synchronized increases in $PM_{2.5}$ concentration along with the development of the mixing layer (MLH < 1200 m) have been witnessed, and the positive response of $PM_{2.5}$ to MLH was significantly associated with the increase in $SO_4^{2-}$ and OC. It was found that this increasing trend of $PM_{2.5}$ with elevated MLH was not only determined by the effect of wet deposition process but also by the enhanced secondary chemical formation, which was related to appropriate meteorological conditions (50 % < RH < 70 %) and increased availability of atmospheric oxidants. Air temperature played a minor role in the change characteristics of $PM_{2.5}$ concentration, but greatly controlled the opposite change characteristics of $SO_4^{2-}$ and $NO_3^-$. The concentrations of $PM_{2.5}$ and its major secondary components, as well as SOR and NOR, increased synchronously with elevated MDA8 $O_3$ concentration, and the initial increase of $PM_{2.5}$ along with the increased MLH corresponded well with that of MDA8 $O_3$. We highlight that the correlation between MLH and secondary air pollutants should be treated with care in hot seasons, and the superposition-composite effects of $PM_{2.5}$ and $O_3$ along with the evolution of mixing layer should be considered when developing $PM_{2.5}$-$O_3$ coordinated control strategies.

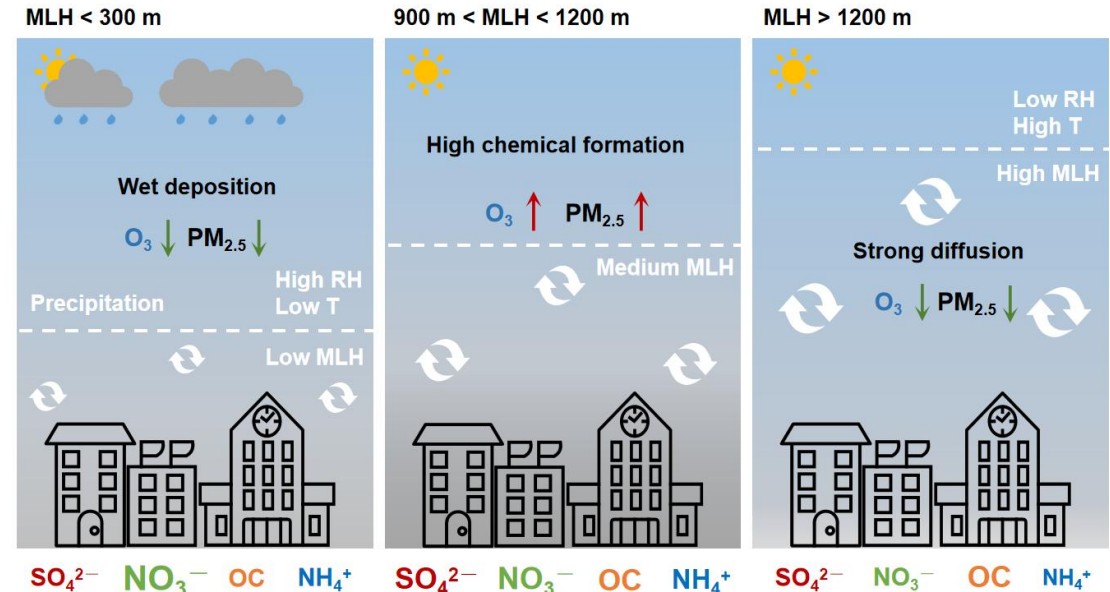

## 1 Introduction

Surface ozone ($O_3$) and $PM_{2.5}$ (atmospheric fine particles with an aerodynamic diameter of less than 2.5 μm) are important air pollutants in the atmosphere and have aroused a lot of attention from the public due to their adverse health impact (Jiang et al., 2018; Cohen et al., 2017; Gao and Ji, 2018). Even though stringent clean air actions have been implemented in China during the past decade, high concentrations of $O_3$ and/or $PM_{2.5}$ exceeding the national air quality standards still occurred during warm seasons (June–August) in China, especially in the North China Plain (NCP), the economic center of China (Dai et al., 2023). $O_3$ is a secondary pollutant originated from photochemical oxidation of volatile organic compounds (VOC) and carbon monoxide (CO) in the presence of nitrogen oxides ($NO_X$), and $PM_{2.5}$ is mainly determined by the atmospheric processes of emissions and secondary formation from gaseous precursors. In addition to air pollutant emissions, meteorological conditions play critical roles in the formation of $PM_{2.5}$ and $O_3$ (Miao et al., 2021). Mixing layer height (MLH), which influences the vertical mixing within the pollution mixing layer and determines the dilution of pollutants emitted near the ground (Haman et al., 2014; Zhu et al., 2018; Lou et al., 2019), often serves as a critical physical parameter in atmospheric environmental evaluation. Elucidating the association of MLH with surface $O_3$ and $PM_{2.5}$ is fundamental for the development of $PM_{2.5}$-$O_3$ coordinated control strategies.

The response of air pollution to MLH was changeable and complicated (Miao et al., 2021). Previous works frequently assumed that the narrowing of mixing layer resulted in accumulation of pollutants near the ground and the increase in MLH was expected to reduce $PM_{2.5}$ concentration due to dilution (Murthy et al., 2020; Du et al., 2013). However, the relationship between mixing layer structure and $PM_{2.5}$ concentration depends on the site, observation period, and the properties of MLH retrievals (Geiß et al., 2017; Lu et al., 2019). Even though the link between $PM_{2.5}$ concentrations and MLH has been investigated in many studies, most observations were conducted in winter conditions and few studies in hot seasons. Interestingly, in some cities, such as Delhi (Murthy et al., 2020) and Shanghai (Pan et al., 2019; Miao et al., 2021), the increase in $PM_{2.5}$ has been observed when MLH increased during summertime. As for $O_3$, the relationship between the changes in the MLH and $O_3$ concentrations is very complex. Both increase or decrease of $O_3$ has been observed corresponded to the growth of MLH. First, $O_3$ concentration decreases along with the increase of MLH owing to dilution.

Second, an increase in MLH generally promotes the downward mixing of upper air containing higher
$O_3$ (Ma et al., 2021; Haman et al., 2014; Xu et al., 2018). In addition, the meteorological conditions
along with the changes of MLH can influence $O_3$ concentrations through effecting $O_3$ gaseous
precursors or production rates (Porter and Heald, 2019; Zhang et al., 2022). The combined effects of
these processes ultimately determine whether $O_3$ decreases or increases.
Other meteorological variables in the mixing layer were also found to significantly affect $PM_{2.5}$
and $O_3$ concentrations. The poor air quality in the NCP was tightly associated with near-surface
southerly winds and warm stagnant conditions during summertime (Zhang et al., 2015a). The increase
in $PM_{2.5}$ concentration often coincided with high relative humidity (RH) conditions (Liu et al., 2017b),
which was beneficial to liquid-phase heterogeneous reactions and fine particle hygroscopic growth
(Seinfeld and Pandis, 2006; Wang et al., 2016; Zhang et al., 2015b). Temperature was essential to
secondary chemical reaction (Dawson et al., 2007). The increase in temperature can promote chemical
reaction rates, but also stimulate the evaporation of semi-volatile aerosol components, such as nitrate
(Wen et al., 2018). For $O_3$, elevated $O_3$ concentrations generally happened on days with strong sunlight
and low wind speeds, which favored the photochemical production and the accumulation of $O_3$ and its
precursors. Several studies have shown that $O_3$ was significantly positive correlated with temperature,
but negatively correlated with RH (Li et al., 2021; Hou and Wu, 2016; Steiner et al., 2010).
Long-term $PM_{2.5}$ composition measurements in the NCP showed an increase in the contributions
of secondary species, e.g., sulfate ($SO_4^{2-}$), nitrate ($NO_3^-$), ammonium ($NH_4^+$), and organic matter (OM),
to total $PM_{2.5}$ in recent years (Cheng et al., 2019; Wang et al., 2022b). As air quality improved ($PM_{2.5}<$
$50 \ \mu g \ m^{-3}$), the correlation between $O_3$ and $PM_{2.5}$ tended to change from negative to positive in China
(Chu et al., 2020). One speculative reason for this phenomenon is that $PM_{2.5}$ does not reduce actinic
flux and $HO_2$ radical significantly when the $PM_{2.5}$ concentration was low. On the other hand, $PM_{2.5}$ and
$O_3$ tend to be positively correlated possibly due to their common precursors, such as VOCs and NOx,
and their simultaneous generation in photochemical reactions. In addition, the generation of $O_3$ would
cause the enhancement of atmospheric oxidation capacity, and catalyze the generation of the secondary
$PM_{2.5}$ (Cheng et al., 2019; Kang et al., 2021; Wu et al., 2022). Even though some studies have
discussed the correlations between MLH and some secondary pollutants, the understanding of the
interaction between $O_3$ and $PM_{2.5}$ (including its major components) along with the evolution of mixing
layer during warm seasons remained poor owing to the limited observations of $PM_{2.5}$ chemical species
involved. The regional-scale observation can represent the variation characteristics for this area and
avoid the spatial heterogeneity between the sites. However, to the best of our knowledge, previous
observational studies were mostly limited to specific cities. Therefore, it's encouraged to analyze
multiple data sources to determine overall trends rather than making conclusions based on a single
dataset.
According to the hourly concentrations of $PM_{2.5}$ and MDA8 $O_3$ in China over the years of
2013–2020, the months of June and July can well represent the typical characteristics of $O_3$–$PM_{2.5}$
coordinated pollution during warm seasons in the North China Plain (NCP) (Dai et al., 2023). To
enhance the understanding of the linkages between mixing layer structure and air pollution, in this
study, a regional-scale field observation of meteorological factors, $O_3$, $PM_{2.5}$ concentration and its
secondary composition were conducted in the NCP from June 1 to July 31, 2021. For the first time, the
potential association among ground-level observed $O_3$, $PM_{2.5}$ and its dominant components, and mixing
layer meteorological conditions will be explored in the NCP during summertime.
**2 Data and methods**
**2.1 Measurements**

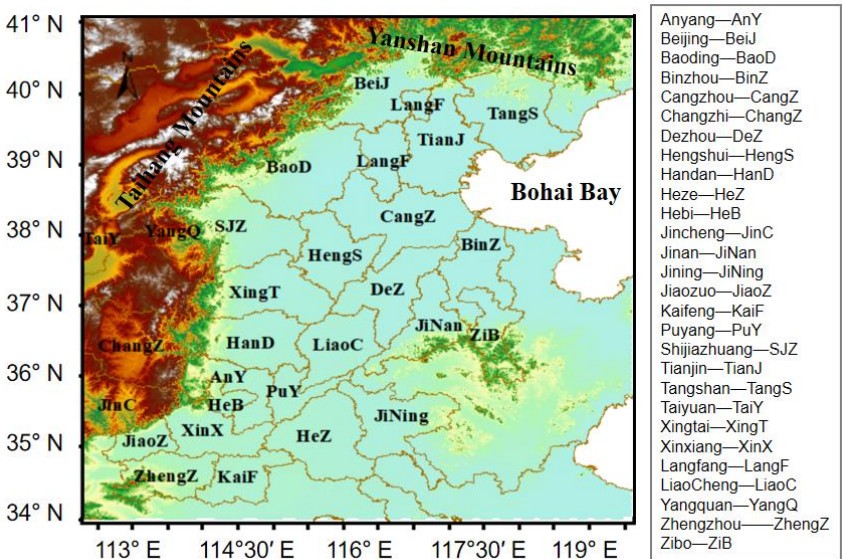


**Figure 1.** Location of monitoring stations in the NCP.
In this study, observation was made in the North China Plain (NCP) from June 1 to July 31, 2021.
The air pollution observation stations in the NCP covered two megacities (BeiJ and TianJ) and 26
surrounding cities. The geographic locations of these stations are marked in Figure 1. The North China
Plain (NCP) is surrounded to the west by the Taihang Mountains, to the north by the Yan Mountains,
and to the east by the Bohai Sea. The hourly concentrations of ground-level $O_3$, $PM_{2.5}$ and its major
components ($SO_4^{2-}$, $NO_3^-$, $NH_4^+$, and OC), and meteorological variables, including air temperature,
relative humidity (RH), wind speed (WS) and direction (WD), and 24-h accumulated precipitation, at
the sites were obtained from the platform of National Atmospheric Particulate
Chemical-Speciation-Network, which is established for improving the understanding of the heavy
pollution formation mechanism in the NCP and supporting the decision-making of local governments
and state administration. Hourly $SO_2$, $NO_2$, $O_3$, $PM_{2.5}$ and its chemical compositions were recorded at
the $PM_{2.5}$ component network, which was selected followed the Technical Regulation for Selection of
Ambient Air Quality Monitoring Station published by the Ministry of Ecology and Environment of the
People's Republic of China (HJ664−2013). The monitoring sites of $PM_{2.5}$ component network were
mostly set up within the cities, and can reflect the average pollution level of each city. Details for the
near-ground observation stations of $PM_{2.5}$ component network were shown in Table S1. Mass
concentrations of $SO_4^{2-}$, $NO_3^-$, and $NH_4^+$ in $PM_{2.5}$ were continuously measured at a 1-h resolution by
MARGA (model ADI 2080) or AIM-IC (URG 9000D) equipped with a $PM_{2.5}$ sampling inlet. These
two IC-based online instruments have shown good performance through instrument intercomparison
studies or comparison to offline filters under clean to moderately polluted conditions (Markovic et al.,
2012; Wu and Wang, 2007; Park et al., 2013; Rumsey et al., 2014). Organic carbon (OC) was measured
online by Sunset Semi-Continuous Carbon Analyzer (Sunset Laboratory Inc, USA). The concentration
of OM can be obtained by multiplying the OC concentration by a factor of 1.6 (Li et al., 2021). $PM_{2.5}$,
$O_3$, $NO_2$ and $SO_2$, were recorded hourly based on Thermo Scientific samplers and analyzers. Detailed
descriptions of these online sampling instruments can be found in our previous works (Kong et al.,
2018; Liu et al., 2017a; Pang et al., 2020; Wang et al., 2022b). The meteorological variables were
recorded in the national meteorological observation stations, and the information of each station can be
obtained from the public website of China Meteorological Administration
(http://data.cma.cn/data/cdcindex/cid/0b9164954813c573.html). The temporal resolution of air
temperature, RH, WS and WD was 1-h. To avoid the influence of diurnal boundary layer cycles, in this
article we focused on the relationships between daily mean air pollutants and meteorological factors.
The daily mean meteorological factors, $PM_{2.5}$ and its major secondary components were calculated
from the hourly data; daily $O_3$ concentration was characterized by the maximum daily 8 h average
ozone (MDA8 $O_3$). Details for the near-ground observation species and the metrics were shown in
Table S2.

To better demonstrate the overall change characteristics of regional air pollution and

meteorological conditions during the observation period, the occurrence frequency (%), which means
the proportion of the number of cities at each air pollutant or methodology level, was calculated based
on the following equation:
$$\text{Occurrence frequency}_X^{\text{level}} = \frac{N_X^{\text{level}}}{\text{Total } N_X} \times 100 \text{ \%} \tag{1}$$
where X means the air pollutants or methodology factors, $N_X^{\text{level}}$ represents the number of cities at each
X level, $\text{Total } N_X$ represents the total number of cities.
**2.2 The calculation of mixing layer heights**

In recent years, many works have progressed in the atmospheric boundary layer characteristics,

and analyzed the impacts of these parameter on air pollution. Planetary boundary layer (PBL), as one
of the critical parameters to air quality modeling, has been well explored. However, PBL usually refers
to the large-scale Ekman dynamic boundary layer (Haugen et al., 1971; Wang et al., 2014; Zhang et al.,
2005). The way with which boundary layer describes the influences of air pollution is easily duplicated
and confused (Niu et al., 2017). It is unreasonable to some extent, if the characteristic of the air
pollution related to near-surface boundary layer is evaluated by using the concept of PBL. For air
pollution measurement, one of selected functionalities of parameterization scheme for pollution mixing
layer is to judge whether an air mass over a specific locality satisfies the "static and stable" attribute or
not. Therefore, in this work, to express the basic physics for diagnosing meteorological conditions, we
used the concept of pollution mixing layer height (MLH) proposed by Wang et al. (2017), which was
based on the classical synoptic theory according to the level of convective condensation layer, and the
details of this method can be seen in previous work (Wang and Yang, 2000; Wang et al., 2017).

To be specific, we define the height close to the cloud base as the height of super-saturation layer

(H_SSL), and the isoentropic atmospheric process meets the level of convective condensation layer
(LCL) in the super-saturation state, i.e., it is very close to the H_SSL. Iterative algorithm is used to
work out the H_SSL (Wang and Yang, 2000):
$$H\_SSL \approx LCL = 6.11 \times 10^2 \times \left( \frac{0.622 + 0.622 \frac{e_s}{p - e_s}}{0.622 \frac{e_s}{p - e_s}} \right), \quad (2)$$
$$e_s = 6.22 \times \exp \frac{17.13(T - 273.16)}{T - 38}, \quad (3)$$
where $e_s$ represents saturated water vapor pressure, T is temperature (K). Eq. (2) can be used to
calculate the H_SSL which is favorable for pollutant mixing and represented by (P). Below this height,
the atmosphere gets supersaturated, causing the pollution mixing and wetting process in the low
altitude to continue, so this height is called the height of pollution mixing layer (MLH). Thus, MLH
can be derived in the following expression:
$$MLH \approx H\_SSL \approx LCL = 6.11 \times 10^2 \times \left( \frac{0.622 + 0.622 \frac{e_s}{p - e_s}}{0.622 \frac{e_s}{p - e_s}} \right), \quad (4)$$
According to the relationship between air pressure and height, the units of MLH can be converted to
the height expression in meters:
$$\int_{p_0}^{p_z} dp = -\int_0^z \rho_0 g dz, \quad (5)$$
where z is the height, $\rho_0$ is the density of gas, $p_z$ and $p_0$ represent the air pressure in the height of z and
0, respectively.
Several works have verified the reliability of the results based on this method. With this method,
Wang et al. (2017) well characterized the features of mixing layer height in highly-sensitive areas of
pollution in China. Wang et al. (2022c) also used this method to explore the $PM_{2.5}$ and $O_3$
superposition-composite pollution event during spring 2020 in Beijing, China, and the hourly evolution
of MLH, $O_3$, and $PM_{2.5}$ during the observation period were analyzed. In addition, Niu et al. (2017) has
applied this method in Beijing, and the results showed that the pollution mixing layer can well present
the change characteristics of haze pollution process. In this work, we further clarified the concept of
MLH, and applied this method to investigate the impacts of MLH upon the change characteristics of
ozone and fine particulate matter.
**3 Results and discussions**
**3.1 General characteristics**

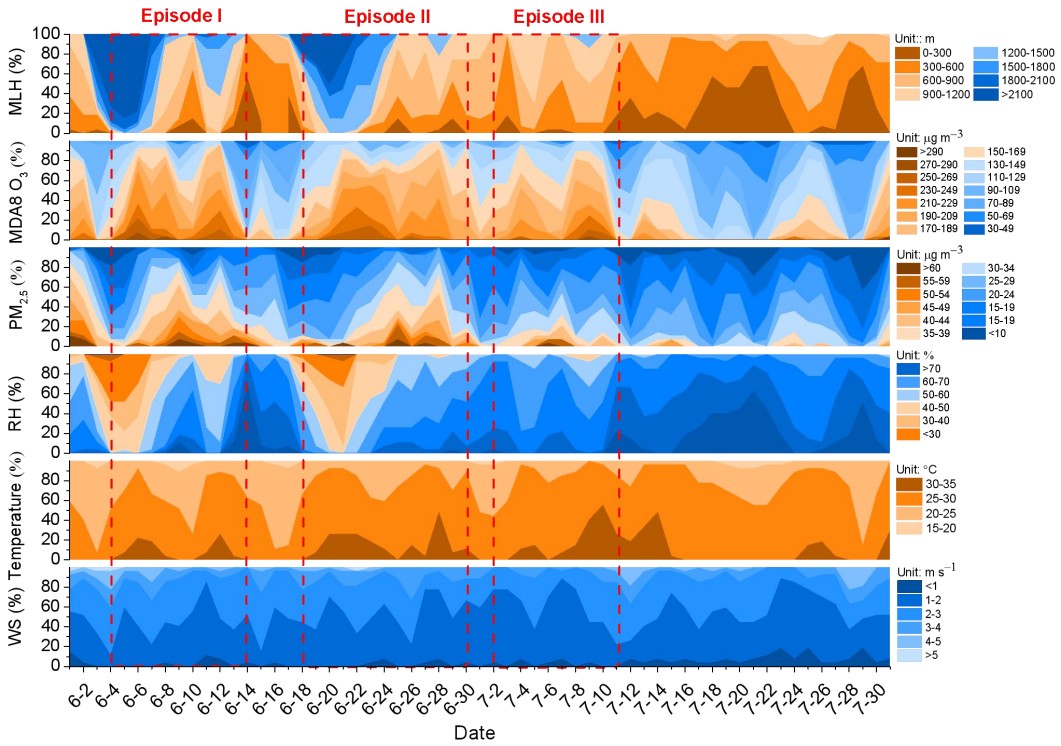


**Figure 2.** The occurrence frequency (%) of PM$_{2.5}$, MAD8 O$_3$, and meteorological factors under
different levels in the NCP from June 1 to July 31, 2021. The color shading represents different
categories classified by PM$_{2.5}$, MDA8 O$_3$, and meteorological factors. The red dash boxes represent
three typical PM$_{2.5}$ and O$_3$ co-polluted episodes: June 4–14 (Episode I), June 18–29 (Episode II), and
July 2–11 (Episode III), 2021.
The summertime change characteristics of ground-level meteorological factors (MLH, RH,
temperature, and WS), MDA8 O$_3$, PM$_{2.5}$ and its major components in the NCP were demonstrated in
Figure 2 and Figure S1. The primary atmospheric pollutant in the NCP during the summertime was O$_3$,
and the concentrations of MDA8 O$_3$ averaged over all sites in the NCP varied from 74.94 μg m$^{-3}$ to
219.28 μg m$^{-3}$, with the mean value of 151.72 μg m$^{-3}$ (Table 1). The O$_3$ pollution lasted nearly the
whole observation period, characterized by frequent and long-lasting pollution episodes. The PM$_{2.5}$
concentration was much lower comparing with ozone, with the mean, maximum, and minimum of the
regional daily mean PM$_{2.5}$ concentration as 25.62 μg m$^{-3}$, 45.62 μg m$^{-3}$, and 11.32 μg m$^{-3}$, respectively,
during the observation period. NO$_3^-$ was the prominent PM$_{2.5}$ component, with the mean concentration
of 7.76 μg m$^{-3}$. According to the National Ambient Air Quality Standard of China (GB3095-2012), the
daily PM$_{2.5}$ averages in "2+26" cities can meet the Level II standard of 75 μg m$^{-3}$, while exceeding the
level I standard (35 μg m$^{-3}$). As showed in Figure 2, regional PM$_{2.5}$ pollution processes corresponded
well with the increasing processes of MDA8 O$_3$. Here, we define a O$_3$–PM$_{2.5}$ co-polluted episode as a
set of continuous days (longer than 4 days) with MDA8 $O_3$ and daily mean $PM_{2.5}$ (in more than 10 %
NCP cities) exceeding 160 µg m$^{-3}$ and 35 µg m$^{-3}$, respectively. According to this criterion, three typical
$O_3$–$PM_{2.5}$ co-polluted episodes were selected: June 4–14 (Episode I), June 18–29 (Episode II), and July
2–11 (Episode III), 2021.
During these three typical episodes, the synchronous change characteristics of air pollutants and
mixing layer meteorology were analyzed. In Episode I and II, when MLH higher than 2100 m, both
MDA8 $O_3$ and $PM_{2.5}$ concentrations were low. Along with the reduction of MLH (from 1800–2100 m
to 1200–1800 m), regional MDA8 $O_3$ and $PM_{2.5}$ concentration both gradually climbed up. When MLH
fell in the range of 1200–1800 m, MDA8 $O_3$ concentration reached the maximum with about 80 %
areas higher than 170 µg m$^{-3}$. We found that there is a lag time between the concentration peak of
MDA8 $O_3$ and that of $PM_{2.5}$ along with the reduction of MLH. With the further decrease of MLH,
MDA8 $O_3$ turned to decline, while $PM_{2.5}$ kept stable or continued to increase when regional MLH in
the range of 600–1200 m. In Episode III, the MLH in most cities was lower than 1200 m, and the
regional MDA8 $O_3$ and $PM_{2.5}$ pollution conditions were lighter than other episodes, with 80 % $PM_{2.5}$
values lower than 35 µg m$^{-3}$. It's interesting to note that the change characteristics of $SO_4^{2-}$ and $NO_3^-$
were different (Figure S1), and the regional peaks of these two components were inconsistent,
especially in Episode II. With the evolution of MLH, $NO_3^-$ climbed up and peaked on June 24 when
regional MLH lower that 900 m, while $SO_4^{2-}$ reached the maximum on June 28 when MLH was around
900–1500 m. This may be related to other synchronized mixing layer meteorology factors, such as RH
and temperature. For example, the evolution of mixing layer often accompanied with changes in
temperature. The increase in temperature can promote the chemical formation rate of these secondary
components, but also stimulate the volatilization of $NO_3^-$ to gaseous state ($HNO_3$) and lead to the
decrease in $NO_3^-$ concentration. Further analysis about the response of $O_3$, $PM_{2.5}$ and its secondary
components to different mixing layer meteorology factors will be conducted in the following sections.
Table 1. General information on $O_3$–$PM_{2.5}$ co-polluted episodes from June 1 to July 31, 2021.

| | Episode I | | | Episode II | | | Episode III | | | Summer | | |
|---|---|---|---|---|---|---|---|---|---|---|---|---|
| | Ave. | Min | Max | Ave. | Min | Max | Ave. | Min | Max | Ave. | Min | Max |
| Gaseous pollutants (µg m$^{-3}$) | | | | | | | | | | | | |
| MDA8 $O_3$ | 170.80 | 85.62 | 219.28 | 180.65 | 142.10 | 204.15 | 168.70 | 111.79 | 199.39 | 151.72 | 74.94 | 219.28 |
| $SO_2$ | 10.01 | 6.48 | 14.44 | 9.09 | 6.11 | 12.48 | 6.75 | 5.72 | 8.00 | 7.59 | 4.79 | 14.44 |

| NO$_2$ | 24.61 | 16.26 | 31.81 | 22.89 | 14.11 | 32.15 | 17.66 | 13.12 | 21.00 | 19.31 | 10.90 | 32.15 |
|---|---|---|---|---|---|---|---|---|---|---|---|---|
| PM$_{2.5}$ and its major components (µg m$^{-3}$) | | | | | | | | | | | | |
| PM$_{2.5}$ | 30.55 | 15.74 | 42.67 | 28.33 | 17.22 | 42.52 | 25.05 | 20.84 | 31.75 | 25.62 | 11.32 | 45.62 |
| NO$_3^-$ | 8.74 | 2.16 | 16.44 | 8.29 | 2.85 | 18.00 | 7.67 | 5.87 | 13.44 | 7.76 | 2.16 | 18.24 |
| SO$_4^{2-}$ | 7.22 | 2.81 | 10.25 | 7.32 | 4.02 | 12.15 | 7.12 | 5.48 | 8.92 | 7.04 | 2.81 | 12.15 |
| NH$_4^+$ | 5.51 | 1.42 | 9.34 | 5.52 | 2.27 | 9.29 | 5.38 | 4.46 | 8.21 | 5.30 | 1.42 | 9.88 |
| OC | 5.11 | 2.74 | 6.60 | 4.71 | 3.25 | 6.75 | 4.11 | 2.90 | 5.30 | 4.32 | 2.69 | 6.75 |
| Meteorological variables | | | | | | | | | | | | |
| MLH (m) | 1342.73 | 305.93 | 2423.42 | 1190.36 | 626.51 | 2127.31 | 740.86 | 460.91 | 950.10 | 855.99 | 305.93 | 2423.42 |
| T (°C) | 26.24 | 23.86 | 28.91 | 27.41 | 25.53 | 28.76 | 27.58 | 24.85 | 30.14 | 26.69 | 22.48 | 30.14 |
| RH (%) | 57.01 | 32.78 | 90.54 | 56.90 | 37.04 | 70.60 | 71.45 | 64.64 | 80.38 | 68.70 | 32.78 | 90.54 |

## 3.2 Evolution of ozone with mixing layer meteorology

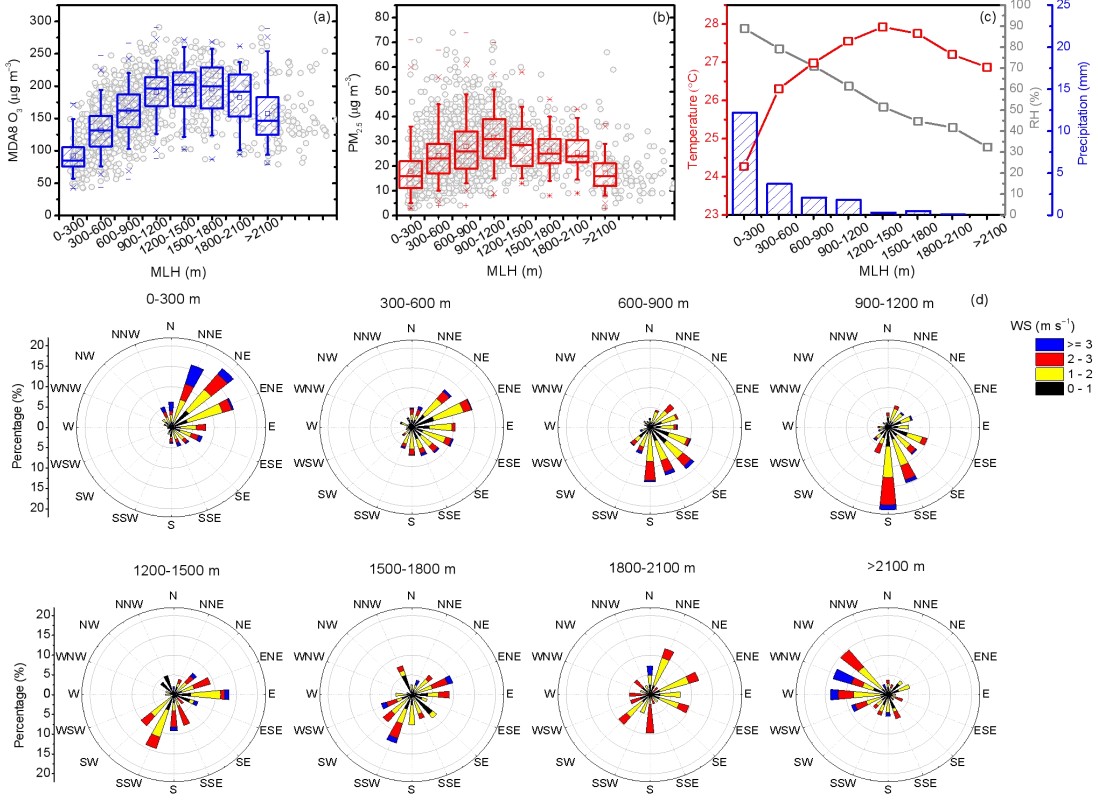

**Figure 3.** The variation characteristics of (a) MDA8 O$_3$, (b) PM$_{2.5}$, (c) temperature, RH, 24-h precipitation, and (d) WS and WD in different MLH conditions. Box plots in (a) and (b) show the inter quartile range (the distance between the bottom and the top of the box), median (the band inside the box), and 95 % confidence interval (whiskers above and below the box) of the data. S: south; N: north; E: east; W: west.

To quantify the effect of MLH on near-ground O$_3$ concentrations, relationships between MLH and

MDA8 O$_3$ were analyzed (Figure 3a). Here we used a data binning method to remove the expected
day-to-day atmospheric variability from sampling uncertainty (Dian et al., 2010), which has been
applied elsewhere (Lou et al., 2019). The MLH was grouped into 8 classes with 300 m width: 0–300,
300–600, 600–900, 900–1200, 1200–1500, 1500–1800, 1800–2100 and > 2100 m. It was found that
MDA8 O$_3$ concentration dramatically increased when MLH in the range of 0–900 m, and leveled off
when MLH at around 900–1800 m, with the maximum MDA8 O$_3$ of 195.88±42.76 μg m$^{-3}$, then turned
to decrease with further development of MLH. This nonlinear relationship between MDA8 O$_3$ and
MLH was is consistent with the results conducted by Zhao et al. (2019) and Reddy et al. (2012). The
work by Zhao et al. (2019) found that O$_3$ concentration was the highest at medium boundary layer
heights (1200–1500 m) in Shijiazhuang, China. In India, days of higher O$_3$ concentrations were also
associated with higher boundary layer height (Reddy et al., 2012).
This relationship observed between MDA8 O$_3$ and MLH is very complex. Previous works have
shown that higher height of mixing layer can lead to the mixing of near surface air with the O$_3$ rich air
aloft, resulting in the observed enhancements in surface O$_3$ concentration (Reddy et al., 2012).
Concurrently, the evolution of mixing layer were strongly associated with the change of other
meteorological conditions, such as air temperature, RH and precipitation, which can also affect O$_3$
concentration (Haman et al., 2014). The combined effects of these processes ultimately determine
whether ground-level O$_3$ increases or not along the evolution of mixing layer. The increase of MLH
often coincided with higher air temperature, lower RH, and less precipitation (Figure 3c), which were
more conducive to O$_3$ production (Ma et al., 2021; Xu et al., 2018). As shown in Figure 4a–c, as MLH
kept constant, MDA8 O$_3$ concentration climbed up with increase in temperature but decrease in RH
and precipitation levels. Possible reasons for these results could be: (1) the increase in RH can
contribute to the depletion of O$_3$, and lead to weakened O$_3$ related photochemical reaction (Ma et al.,
2021; Yu, 2019); (2) due to higher RH or rain fall, gaseous precursors and O$_3$ can be washed out from
the atmosphere trough wet deposition (Reddy et al., 2012); and (3) the rise of temperature can
accelerate the emission rate of gaseous precursors, such as biogenic VOCs and soil NO$_X$ (Dang et al.,
2021; Porter and Heald, 2019), and also stimulate the photochemical reaction rate in the generation of
O$_3$ (Ma et al., 2021). Besides, wind fields could alter surface O$_3$ concentrations by transporting O$_3$ or its
precursors in and out of this region (Ma et al., 2021). As shown in Figure S2, during the whole
campaign, the NCP was dominated by winds from northeast and south (45−225°). Because more than
75 % WD were in the rang of 45−225°, the WD was classified into 4 categories: 45−90, 90−135,
135−180, and 180−225°. As shown in Figure 3d, along with the evolution of mixing layer, the WD
gradually changed from northeast (MLH=0−600 m) to southeast (MLH=600−900 m) and south
(MLH=900−1200 m). The southerly wind can transport the gaseous pollutants or $O_3$ from the southern
part of the plain area to the northern part, and the Taihang mountains may block pollutant transport,
leading to the accumulation of pollutants along the foot of the Taihang Mountains. It's noted that the
concentration of MDA8 $O_3$ was higher when the plain dominated by southerlies (180−225°) when
MLH lower than 1200 m (Figure 4e). In general, WS could affect the diffusion of air pollutants. Due to
the limited dilution and dispersion effect of weak wind, the MDA8 $O_3$ concentrations at low wind speed
(0−1 m s$^{-1}$) were relative higher than other WS conditions (Figure 4d) .

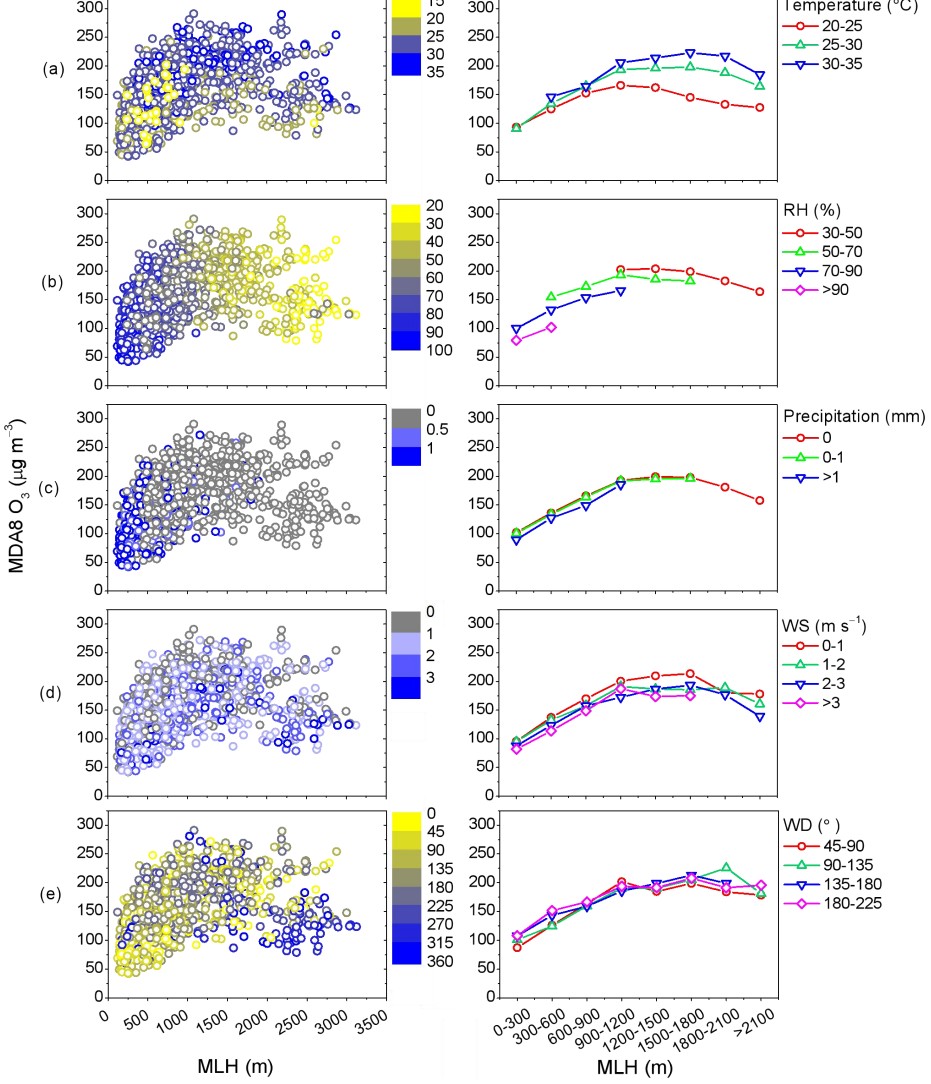

**Figure 4.** The distribution characteristics of the MDA8 O$_3$ concentrations with the evolution of MLH
under different (a) temperature, (b) RH, (c) precipitation, (d) WS, and (e) WD conditions.
**3.3 Evolution of PM$_{2.5}$ and its secondary compositions with mixing layer meteorology**
The concentration distribution of surface PM$_{2.5}$ in different MLH bins has been shown in Figure
3b. Interestingly, PM$_{2.5}$ showed similar change profile as MDA8 O$_3$, which initially increased and then
declined along with the growth of MLH. PM$_{2.5}$ concentration reached the maximum of 31.65 μg m$^{-3}$
when MLH fell in the range of 900–1200 m, and the concentration has increased by 1.51 μg m$^{-3}$
through the rise phase for the variation of 100 m MLH. This phenomenon was quite different with the
results in cold seasons (Pan et al., 2019; Du et al., 2013; Murthy et al., 2020). It has been suggested that
the narrowing of mixing layer will compress air pollutants into a shallow layer, resulting in elevated
pollution levels, thus MLH has been illustrated as the key factor to aggravate the haze events in large
cities of China in winter. However, the response of PM$_{2.5}$ concentration to MLH is not only determined
by the vertical stratification of the mixing layer, but also by local sources, secondary chemical
formation, wet deposition, and the wind field (Lu et al., 2019; Geiß et al., 2017; Pan et al., 2019; Miao
et al., 2021; Lou et al., 2019). Noted that in this work there were still some extreme high PM$_{2.5}$ values
under low MLH condition as shown in Figure 3b, and this phenomenon will be discussed in the
following part when exploring the effect of precipitation.

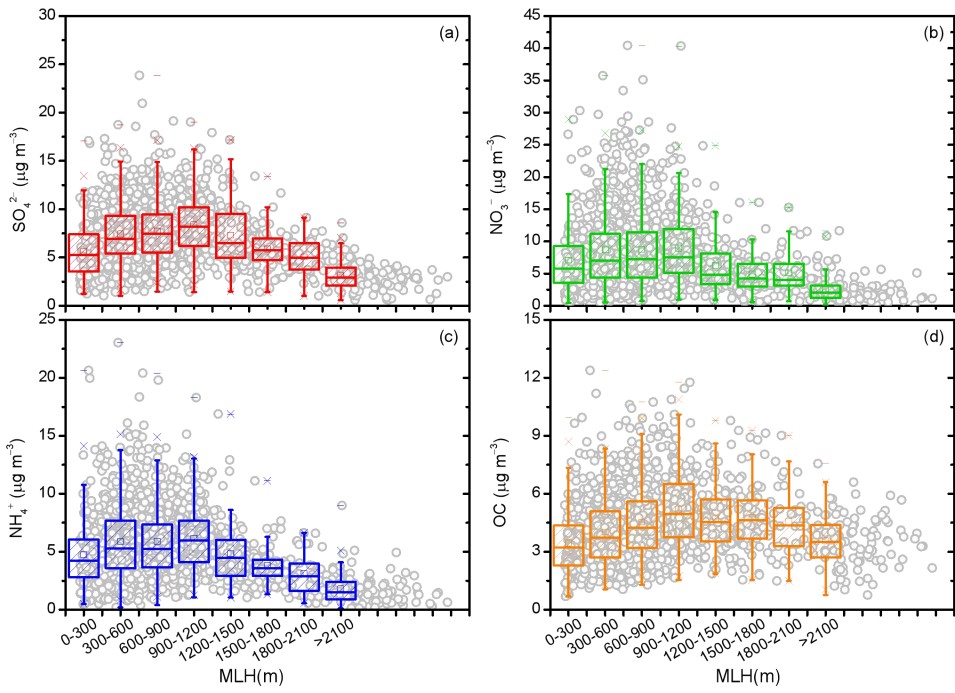


Figure 5. The variation characteristics of (a) SO$_4^{2-}$, (b) NO$_3^-$, (c) NH$_4^+$, and (d) OC in different MLH
conditions. Box plots show the inter quartile range (the distance between the bottom and the top of the
box), median (the band inside the box), and 95 % confidence interval (whiskers above and below the
box) of the data.

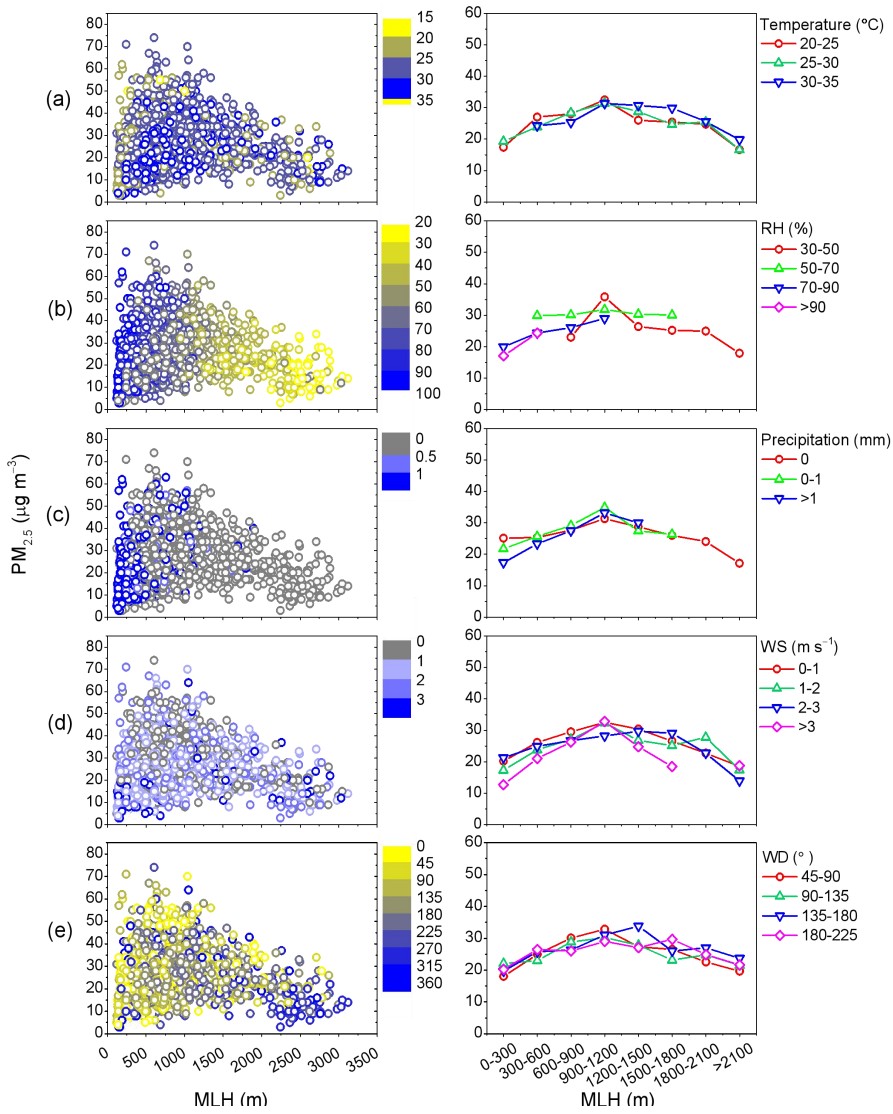

**Figure 6.** The distribution characteristics of the PM$_{2.5}$ concentrations with the evolution of MLH under
different (a) temperature, (b) RH, (c) precipitation, (d) WS, and (e) WD conditions.

The response of PM$_{2.5}$ concentrations to mixing layer structure was the net effect of the changes in

PM$_{2.5}$ major chemical components, such as SO$_4^{2-}$, NO$_3^-$, NH$_4^+$, and OC. Figure 5 showed the changes
in the major PM$_{2.5}$ components due to the evolution of mixing layer. All the secondary components
showed increasing trends when MLH lower than 1200 m, with SO$_4^{2-}$ and OC showing the highest
increment, followed by NO$_3^-$ and NH$_4^+$. When MLH changed from 300−600 m to 900−1200 m, the
increment was not significant for NO$_3^-$ and NH$_4^+$. As NH$_3$ was generally abundantly supplied in the
NCP, the formation of NH$_4^+$ was dominantly controlled by the reaction of ammonia with sulfate and
nitrate aerosols, and the changes in $NH_4^+$ were a consequence of the changes in $SO_4^{2-}$ and $NO_3^-$ (Chow
et al., 2022). When MLH<1200 m, the mass fraction of $NO_3^-$ was higher than $SO_4^{2-}$ in $PM_{2.5}$ (Figure
S3), and the change characteristics of $NH_4^+$ along with the evolution of mixing layer were consistent
with that of $NO_3^-$. The mass ratio of $SO_4^{2-}$ to $NO_3^-$ gradually increased along with the development of
mixing layer. When MLH higher than 1200m, $SO_4^{2-}$ surpassed $NO_3^-$ and became the dominant $PM_{2.5}$
component. The difference in the relationships between these aerosol species and MLH reflected the
intrinsic complexity mechanisms of $PM_{2.5}$ formation, which were probably related to other
meteorological parameters, such as temperature, RH, precipitation, WS, and WD. In order to
understand how the other meteorological factors impacted the relationship between MLH and $PM_{2.5}$,
we demonstrated the statistics on the concentration distribution of $PM_{2.5}$ and its dominant components
with the increase of MLH under different RH, temperature, precipitation, WS, and WD conditions in
Figure 6 and Figure 7.
Temperature is not only essential to the secondary chemical reaction of trace gases, but also the
gas–particle partitioning of volatile $PM_{2.5}$ species. The response of $PM_{2.5}$ and its dominant components
to MLH followed similar change characteristics under different temperature conditions, all increasing
with the development of mixing layer when MLH lower than 1200m. The response of $PM_{2.5}$ to
temperature was largely the result of opposite changes in $NO_3^-$ and $SO_4^{2-}$ concentrations with a smaller
role played by organics (Figure 7). Specifically, as MLH kept constant, $SO_4^{2-}$ concentration climbed up
with increasing temperature level, while the concentration of $NO_3^-$ declined when temperature kept
going up. Higher temperature may promote faster oxidation of $SO_2$ to $SO_4^{2-}$, resulting in a significant
increase in $SO_4^{2-}$ concentrations. Unlike $SO_4^{2-}$, which predominantly exists in the particle phase, $NO_3^-$
could be either presented as nitric acid ($HNO_3$) in the gas phase or as ammonium nitrate ($NH_4NO_3$) in
the particle phase (Chow et al., 2022). The temperature condition strongly influences the partitioning of
nitrate between gas and particle phase. Higher temperature prompts the partitioning of nitrate to $HNO_3$,
thus nitrate tends to exit in the gas phase, resulting in a significant decrease in $NO_3^-$ and $NH_4^+$
concentrations.
The response of $PM_{2.5}$ and its dominant components to the evolution of mixing layer was more
sensitive to RH, and distinct distribution characteristics under different RH ranges have been observed
in Figure 6 (b) and Figure 7 (b). When MLH fell in the range of 300–900 m, the concentration of $PM_{2.5}$
(Figure 6b) and its major components (Figure 7b) mostly decreased with RH elevating from 50–70 %

to 70–90 %. Previous works have shown that when RH higher than 60%, local humidity-related physicochemical processes play important roles in transforming the gases to aerosols (Wang et al., 2022d; Liu et al., 2020). We considered that the RH range from 50% to 70% was more beneficial to the aqueous chemical production of major $PM_{2.5}$ components, then led to the increase of $PM_{2.5}$ concentration. It is worth noting that when MLH in the range of 0–300 m, with RH increasing from 70–90 % to > 90 %, the concentration of $PM_{2.5}$ (Figure 6b) and its major components (Figure 7b) severely dropped, which was probably related to the fast hygroscopic growth and enhanced wet deposition processes.

All aerosol species have wet deposition as a major sink, so precipitation is expected to have significant effects on $PM_{2.5}$ concentrations. As shown in Figure 6 (c), changes in the concentrations of $PM_{2.5}$ was sensitive to the rain events. When MLH fell in the range of 0–300 m, the concentration of $PM_{2.5}$ significantly decreased during rainfall period. Interestingly, when no rainfall occurred, even though $PM_{2.5}$ concentration kept stable under low MLH condition, the response of $PM_{2.5}$ concentrations to MLH still followed upward trend with MLH increasing from 300–600 to 900–1200 m. As for specific aerosol species (Figure 7c), $NO_3^-$ and $NH_4^+$ concentration showed two obvious peaks, with one in the range of 0–300 m, and the other in 900–1200 m. Under low MLH condition, the concentrations of $NO_3^-$ and $NH_4^+$ were high, with $NO_3^-$ as the dominant species in $PM_{2.5}$ (Figure 8b). With the growth of MLH, $NO_3^-$ and $NH_4^+$ initially decreased, but turned to increase again when MLH in the range of 900–1200 m. As for $SO_4^{2-}$ and OC, the concentrations obviously increased with the elevation of MLH and has exceeded that of $NO_3^-$ when MLH higher than 1200 m. As shown in Figure 8 (a), low mixing layer generally accompanied with cloudy and rainy conditions during summertime in the NCP in 2021, and only small fraction of days without rainfall has been captured during this period. Therefore, despite some high $PM_{2.5}$ or major aerosol species values have been witnessed under low MLH condition, the overall trend in Figure 3 (b) was still upwards along with the growth of mixing layer (MLH < 1200 m). The increase of $PM_{2.5}$ and its major chemical components under medium MLH condition was not only associated with the weaker particle removal process by precipitation, but also related to the enhancement of secondary aerosol formation due to appropriate chemical reaction environment.

WS can represent the atmospheric dissipation potential in the horizontal directions (Zhu et al., 2018). Low WS generally suggested weak pressure gradients and potentially a more favorable meteorological condition for $PM_{2.5}$ enhancement (Ma et al., 2021). As expected, the concentrations of

PM$_{2.5}$ (Figure 6d) and its aerosol species (Figure 7d) gradually decreased with the increase of WS. The
response of these air pollutants to MLH followed similar upward trends under different WS conditions
(MLH < 1200 m). Comparing with O$_3$, the impact of WD along with the increase of MLH seems
different for PM$_{2.5}$ and its dominant components. When MLH in the range of 600−1200 m, the NCP
was dominated by southeast or south wind (Figure 3d). However, when southeast or south wind
prevailed, the corresponding PM$_{2.5}$ and its dominant components concentrations were comparable or
even lower than other WD situations (Figure 6e and Figure 7e). This indicated that regional transport
was not the dominant factor leading to the elevation of PM$_{2.5}$ and its aerosol species along with the
evolution of mixing layer (MLH < 1200 m).

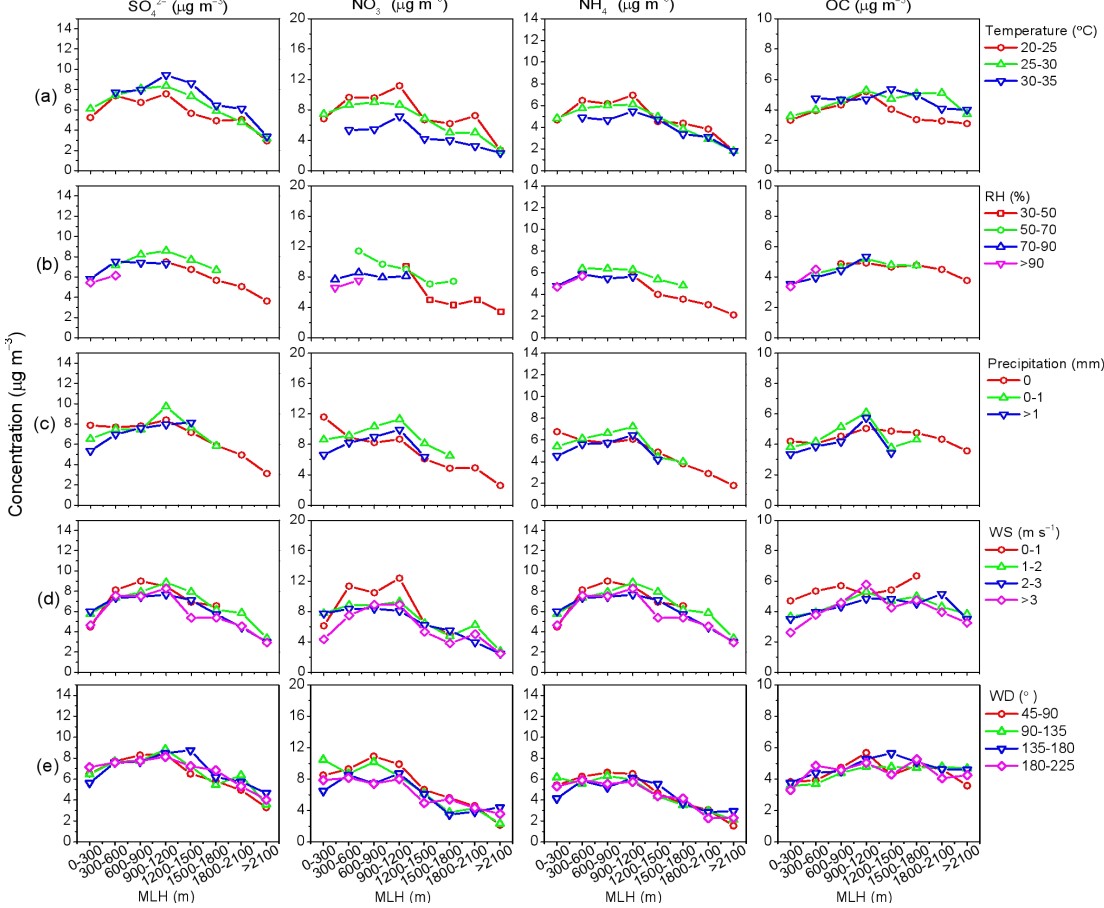

**Figure 7.** The distribution characteristics of NO$_3^-$, SO$_4^{2-}$, NH$_4^+$, and OC concentrations with the
evolution of MLH under different (a) temperature, (b) RH, (c) precipitation, (d) WS, and (e) WD
conditions.

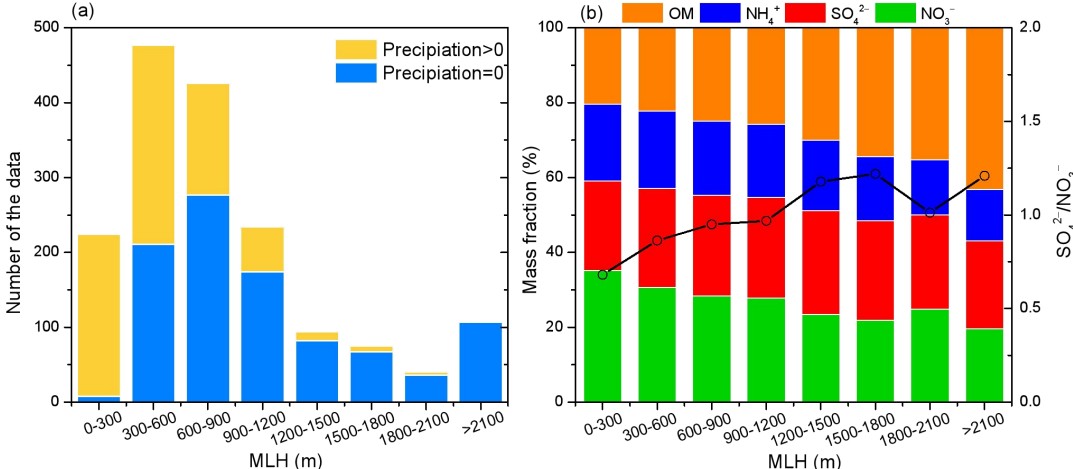


**Figure 8.** (a) The number distributions of the data when the daily precipitation larger than 0 mm or

equal to 0 mm along with the evolution of MLH. (b) The mass fractions of major PM$_{2.5}$ components

and the mass ratio of SO$_4^{2-}$ to NO$_3^-$ along with the evolution of MLH when the daily precipitation

equal to 0 mm.

**3.4 Superposition-composite effects of PM$_{2.5}$ and O$_3$ with the evolution of mixing layer**

**3.4.1 A case study of the typical PM$_{2.5}$-O$_3$ co-polluted episode**

Previous results indicated that MDA8 O$_3$ and PM$_{2.5}$ concentrations were closely related to the

evolution of BLH. The increasing trend of PM$_{2.5}$ concentration with the development of mixing layer

under medium MLH condition discussed before indicated that the evolution of mixing layer was not a

simple physical dilution process, and its influence on the enhanced secondary photochemical formation

should be considered as well. Figure 9 and 10 demonstrated a typical PM$_{2.5}$-O$_3$ co-polluted episode

(Episode II) during 18−29, 2021 to comprehensively present the relationship between the mixing layer

and pollutants. On June 18−20, MLH gradually increased from 600−1200 m to 1500−3000 m in the

southern and eastern part of the NCP, PM$_{2.5}$ and MDA8 O$_3$ concentrations concurrently increased and

showed similar spatial distributions. The wind speed dropped significantly on 20 June, and the value

was lower than 1 m s$^{-1}$ in most cities. On 21−23 June, MLH started to decrease from 1500−3000 m to

1200−1800 m, PM$_{2.5}$ and MDA8 O$_3$ concentrations further increased, and the areas of high PM$_{2.5}$

concentrations also coincided well with those of MDA8 O$_3$ concentrations. During 24−25 June, MLH

continued to decrease, with some values even lower than 300 m. The MLH for the areas with high

MDA8 O$_3$ was in the range of 900−1500 m. Interestingly, the synchronized spatial change

characteristics of PM$_{2.5}$ and MDA8 O$_3$ were consistent when MLH in the range of 900−1200 m, while

inconsistent when MLH lower than 600 m. Significant rise of $PM_{2.5}$ concentration was observed in
some cities with MLH lower than 300 m. It's noted that the dominant chemical composition of $PM_{2.5}$ in
these areas was $NO_3^-$. On 28 June, the rise in MLH was observed in the central and the southern part in
the NCP, and a surge of MDA8 $O_3$ and $PM_{2.5}$ concentrations both occurred, with 160−220 $\mu g\,m^{-3}$ and
40−50 $\mu g\,m^{-3}$ respectively. In general, most cities were dominated by weak winds from the east and
southeast, which favored the formation of secondary pollutants from the gaseous precursors transported
from the southeast part and promoted the accumulation of air pollutants.

To better understand this $PM_{2.5}$-$O_3$ co-polluted event, here we classified the observations during

this typical event into four categories: $O_3$ polluted days ($O_3PD$; MDA8 $O_3$ concentration $> 160\,\mu g\,m^{-3}$
and $PM_{2.5} < 35\,\mu g\,m^{-3}$), $PM_{2.5}$ polluted days ($PM_{2.5}PD$; MDA8 $O_3$ concentration $< 160\,\mu g\,m^{-3}$ and
$PM_{2.5} > 35\,\mu g\,m^{-3}$), $O_3$–$PM_{2.5}$ co-pollution days ($O_3$–$PM_{2.5}CPD$; MDA8 $O_3$ concentration $> 160\,\mu g\,m^{-3}$
and $PM_{2.5} > 35\,\mu g\,m^{-3}$), and non-polluted days (NPD; MDA8 $O_3 < 80\,\mu g\,m^{-3}$ and $PM_{2.5} < 35\,\mu g\,m^{-3}$).
Figure 11 showed the meteorological and chemical characteristic of $O_3$–$PM_{2.5}$ CPD, $O_3PD$, $PM_{2.5}$ PD,
and NPD. The results indicated that the values of MLH on $O_3$–$PM_{2.5}CPD$ were between those on $O_3PD$
and $PM_{2.5}PD$ at around 900 m. On $O_3$–$PM_{2.5}CPD$, the oxidation ratio of sulfate (SOR, the molar ratio
of sulfate to the sum of sulfate and $SO_2$) and oxidation ratio of nitrate (NOR, the molar ratio of nitrate
to the sum of nitrate and $NO_2$) were the highest, with the values of 0.44 and 0.33, respectively, which
indicated the strong secondary formation of $SO_4^{2-}$ and $NO_3^-$ promoted by high $O_3$ concentration. The
$PM_{2.5}PD$ occurred when MLH lower than 650 m, and the percentage of $NO_3^-$ was the highest on
$PM_{2.5}PD$. The rise of $PM_{2.5}$ in some cities under low MLH conditions may be attributed to three
mechanisms. The first one is the accumulation effect due to unfavorable diffusion condition when
MLH decreased. Second, these cities got little rain, and the effect of wet deposition was weak. In
addition, the corresponding low T and high RH can stimulate the formation of $NO_3^-$ from gaseous state
($HNO_3$). On $O_3PD$, the MLH was around 1300 m, and the NOR turned to decrease, demonstrating a
more significant role of partitioning process between gas and aerosol than the atmospheric oxidation
process under this stage. On NPD, the MLH was the highest, with the value of about 2400 m, and the
$PM_{2.5}$ chemical composition was obviously dominated by OM.

To explore the relevance of hourly $O_3$, $PM_{2.5}$, its components and MLH, we have taken PuY and

HeZ as examples. Figure S4 plotted the day-to-day variations along with the diurnal variations of $O_3$,
$PM_{2.5}$, its components and MLH in PuY and HeZ during Episode II (June 18−29, 2021). The results
showed that there were large diurnal as well as day-to-day variability in the $O_3$ and $PM_{2.5}$ levels. The
diurnal variations of MLH were clearly visible (Figure S5), with the rise in MLH during the daytime
and the decrease in MLH at night. The concentration of $PM_{2.5}$ increased with the decrease of MLH at
night, but the concentration of $O_3$ increased with the rise of MLH at daytime. Interestingly, we observed
noontime soar of $SO_4^{2-}$ and OC concentrations in PuY, and the values of SOR kept stable or even
increase at noon. Besides, it's noted that $O_3$ and $PM_{2.5}$ both gradually accumulated with the
development of mixing layer during June 18−21 and 26−28, which can be attributed to the $O_3$ and $PM_{2.5}$
superposition composite effects. The decrease in $PM_{2.5}$ at daytime with the rise of MLH can be offset
partly by an increment in secondary pollutants formation derived from $O_3$ growth. Then with the
decrease of MLH at night, the concentration of the original existing $PM_{2.5}$ increased due to unfavorable
diffusion. In general, the conclusions in this work was only suitable to the day-to-day relationship
between air pollutants and MLH. The hourly relationships were much more complicated and need more
further analysis.


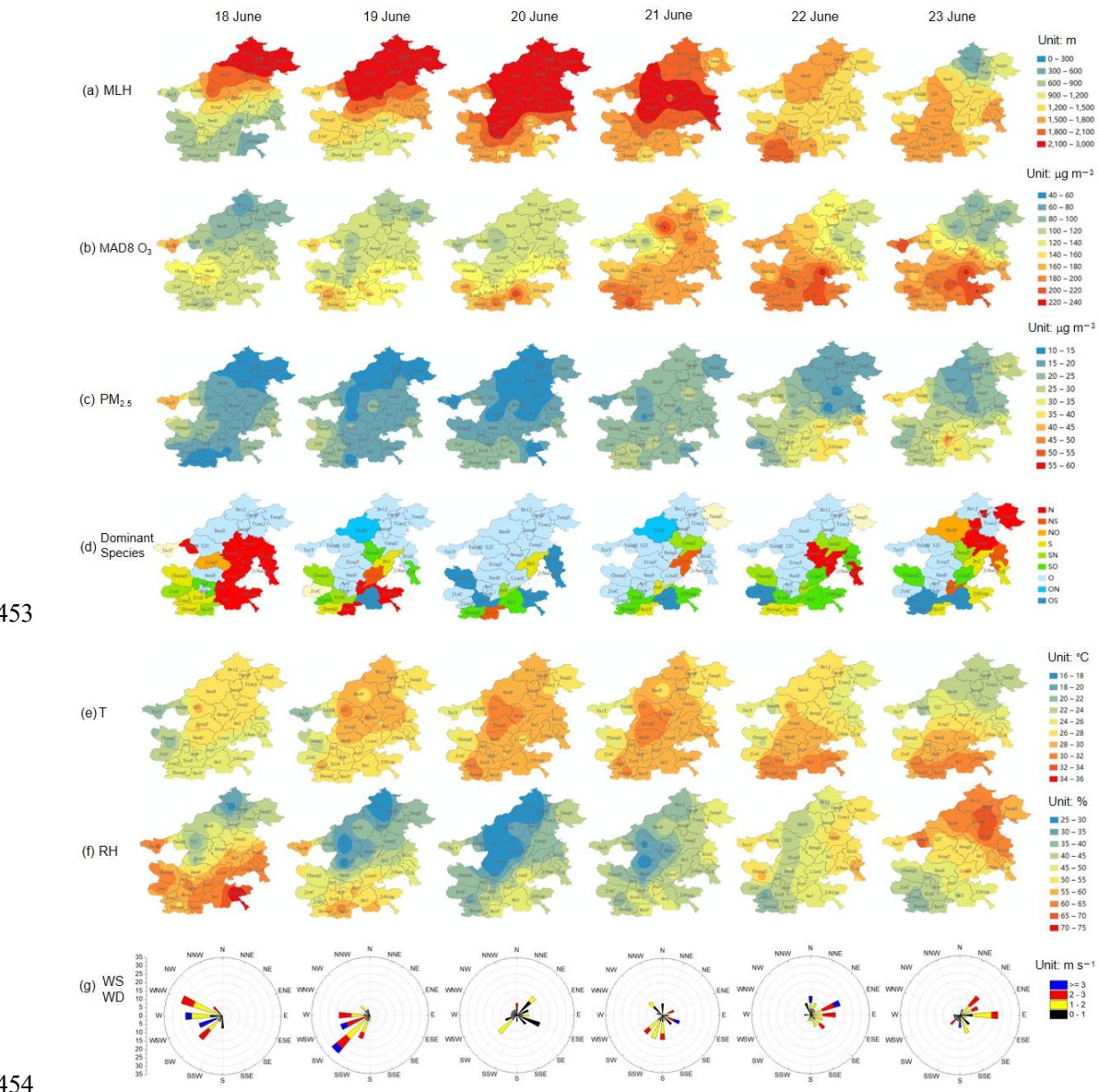


**Figure 9.** The spatial distribution of (a) MLH, (b) MDA8 $O_3$, (c) $PM_{2.5}$, (d) the dominant $PM_{2.5}$ chemical component (N: $NO_3^-$ dominant, NS: $NO_3^-$ and $SO_4^{2-}$ dominant, NO: $NO_3^-$ and OM dominant, S: $SO_4^{2-}$ dominant, SN: $SO_4^{2-}$ and $NO_3^-$ dominant, SO: $SO_4^{2-}$ and OM dominant, O: OM dominant, ON: OM and $NO_3^-$ dominant, OS: OM and $SO_4^{2-}$ dominant), (e) T, and (f) RH, (g) the overall change characteristics of WS and WD in the NCP from June 18 to 23, 2021. The dominant $PM_{2.5}$ chemical component type was identified as the method proposed by Wang et al. (2022b): if the mass fraction of the maximum component was 1.2 times higher than that of the secondary one, the former was considered as the dominant factor, otherwise both dominated $PM_{2.5}$ formation.

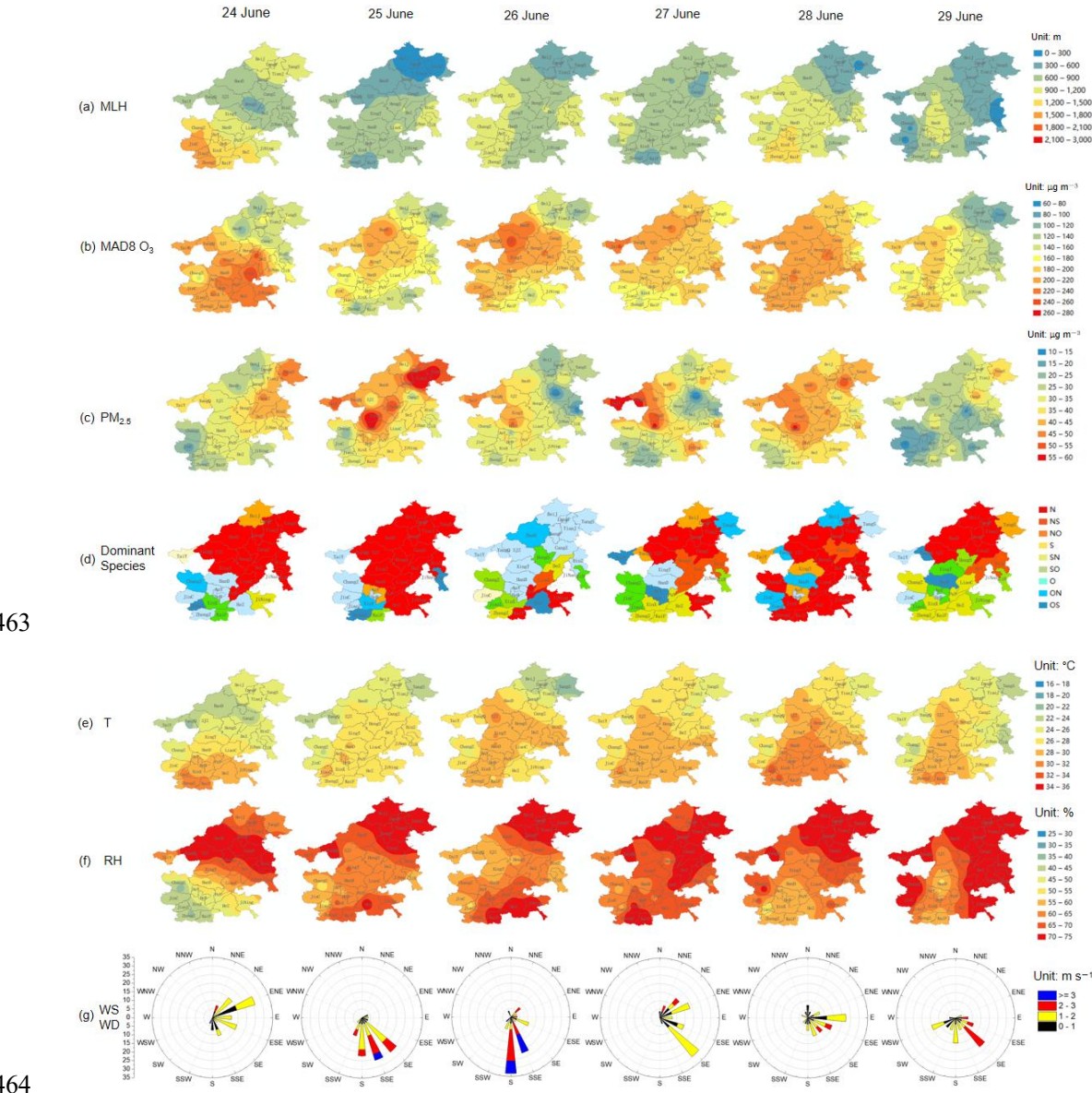



**Figure 10.** The spatial distribution of (a) MLH, (b) MDA8 O₃, (c) PM₂.₅, (d) the dominant PM₂.₅
chemical component (N: $NO_3^-$ dominant, NS: $NO_3^-$ and $SO_4^{2-}$ dominant, NO: $NO_3^-$ and OM dominant,
S: $SO_4^{2-}$ dominant, SN: $SO_4^{2-}$ and $NO_3^-$ dominant, SO: $SO_4^{2-}$ and OM dominant, O: OM dominant, ON:
OM and $NO_3^-$ dominant, OS: OM and $SO_4^{2-}$ dominant), (e) T, and (f) RH, (g) the overall change
characteristics of WS and WD in the NCP from June 24 to 29, 2021. The dominant PM₂.₅ chemical
component type was identified as the method proposed by Wang et al. (2022b): if the mass fraction of
the maximum component was 1.2 times higher than that of the secondary one, the former was
considered as the dominant factor, otherwise both dominated PM₂.₅ formation.

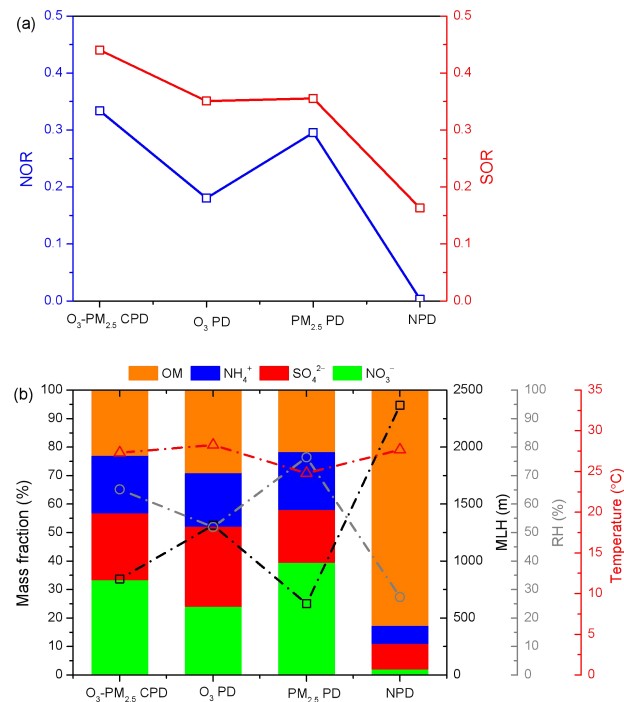


**Figure 11.** The distribution characteristics of (a) NOR and SOR, and (b) the mass fractions of major
$PM_{2.5}$ components, MLH, RH, and temperature under $O_3$–$PM_{2.5}$ CPD, $O_3$ PD, $PM_{2.5}$ PD, and NPD
conditions from June 24 to 29, 2021.
**3.4.2 Interaction between $PM_{2.5}$ and $O_3$ along with the evolution of MLH**

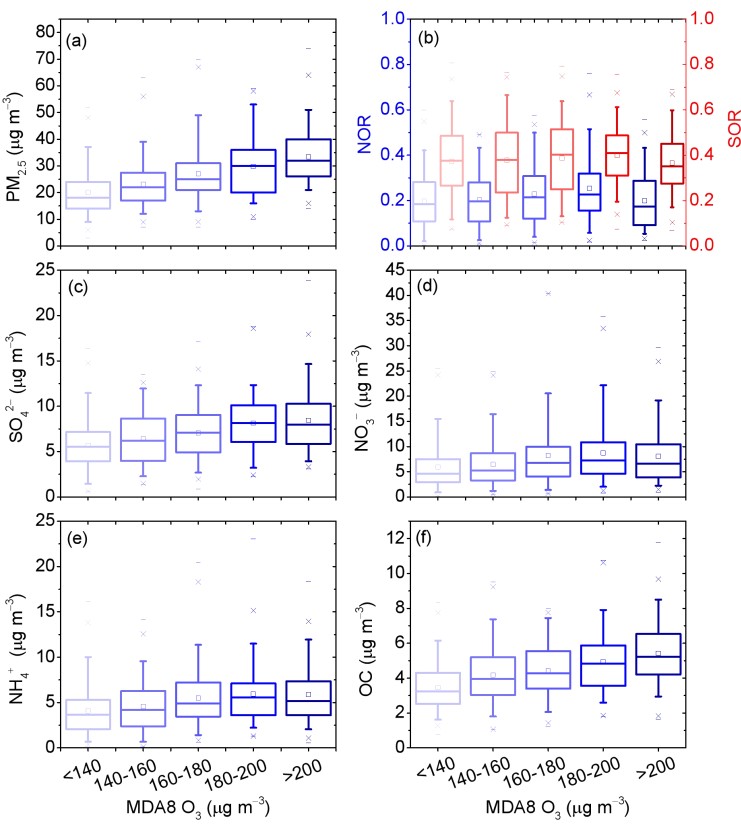


**Figure 12.** Box plots showing the statistics of (a) $PM_{2.5}$, (b) NOR and SOR, (c) $SO_4^{2-}$, (d) $NO_3^-$, (e)
$NH_4^+$, and (f) OC for different MDA8 $O_3$ conditions (< 140 μg m$^{-3}$, 140–160 μg m$^{-3}$, 160–180 μg m$^{-3}$,
180–200 μg m$^{-3}$, > 200 μg m$^{-3}$). The distance between the bottom and the top of the box reflects the
inter quartile range; the line and square in between are the median and mean values, respectively. The
whiskers above and below the box refer the 95 % confidence interval of the data. Note that rainy days
were excluded.

Figure 12 displays the box-and-whisker plots of $PM_{2.5}$ and its major components for different

MDA8 $O_3$ conditions. To isolate the impacts of precipitation on $PM_{2.5}$ concentration, these rainy days
when the daily rainfall amount greater than 0 mm were excluded in this section. Here the
concentrations of $PM_{2.5}$ and its major components were found to increase synchronously with elevated
MDA8 $O_3$ concentration, especially when MDA8 $O_3$ increased from < 140 to 180–200 μg m$^{-3}$.This
summertime collaborative growth process of $PM_{2.5}$-$O_3$ has also been observed in other works (Wang et
al., 2022a; Wu et al., 2022). With elevated MDA8 $O_3$ concentration, SOR and NOR both slightly
increased, and reached the maximum when MDA8 $O_3$ at around 160–200 μg m$^{-3}$, which indicated the
strong secondary formation of $SO_4^{2-}$ and $NO_3^-$ promoted by high $O_3$ concentration. When MDA8 $O_3$
increased from 180–200 to > 200 μg m$^{-3}$, the concentrations of $NO_3^-$, $NH_4^+$, and $SO_4^{2-}$ kept stable or
started to decrease, and the values of SOR and NOR decreased synchronously. During this stage, the
high $O_3$ concentration often accompanied with dry and hot meteorological conditions, which was not
beneficial to the aqueous chemical production and was conducive to the partitioning of nitrate to the
gas phase.

To verify the potential impact of photochemical oxidation to the increase of $PM_{2.5}$ concentration

with mixing layer development, the changes in $PM_{2.5}$ and MDA8 $O_3$ along with the increase of MLH
were quantified in the "2+26" cities in the NCP. Linear regression equations between air pollutants and
MLH were fitted during the initial increasing stage (300 < MLH < 1200 m) and their slopes were given
in Figure 13. The slopes indicated the rates of the maximum changes in air pollutant concentration for a
unit change in MLH (100 m). The slopes of $PM_{2.5}$ and $O_3$ were expressed as $\Delta PM_{2.5}$ and $\Delta O_3$ (μg m$^{-3}$
(100) m$^{-1}$). It was found that $\Delta PM_{2.5}$ was closely related to $\Delta O_3$ ($R^2$=0.58), and obvious spatial
difference in $\Delta PM_{2.5}$ and $\Delta O_3$ was witnessed in the NCP during the observation period. $\Delta PM_{2.5}$ and $\Delta O_3$
both showed high values in YangQ, , LangF and CangZ, with values of 7.56 and 20.24 μg m$^{-3}$ (100)
m$^{-1}$ in YangQ, 5.75 and 18.97 μg m$^{-3}$ (100) m$^{-1}$ in LangF, and 4.02 and 19.49 μg m$^{-3}$ (100) m$^{-1}$ in
CangZ, respectively. Comparing with these cities, $\Delta PM_{2.5}$ and $\Delta O_3$ were lowest in HeB, with the value
of 3.54 and $-2.02$ $\mu g$ $m^{-3} (100)$ $m^{-1}$, respectively, which implied that the secondary formation here was
weak and the surface $PM_{2.5}$ change characteristic was dominantly controlled by local emissions or
vertical diffusion effect.

513   Comparing with winter, the photochemistry in summer is quite active due to the strong solar

radiation. Even though deep MLH favors the dilution of air pollutants, higher MLH can also promote
secondary chemical feedback through enhancing the availability of atmospheric oxidation capacity
(such as changes in $O_3$) along with appropriate meteorological condition. This conclusion corresponded
well to the finding based on chemical transport model (Dai et al., 2023), which proposed strong
chemical production of secondary aerosols when planetary boundary layer height was about 946.1m on
$O_3$–$PM_{2.5}$ co-pollution days. The strong chemical productions in the oxidative atmosphere at medium
MLH condition may overcome the dilution effect on $PM_{2.5}$ induced by mixing layer development,
leading to higher $PM_{2.5}$ level at the ground level. However, it should be noted that the conclusions in
this work were only suitable to summertime regional observations, especially for warm and humid
seasons. The conditions would be different in wintertime (much lower $O_3$ level). More extended
observations in time and space should be needed in the future to further examine and better understand
the complex interactions between MLH, air pollution, and chemical processing.

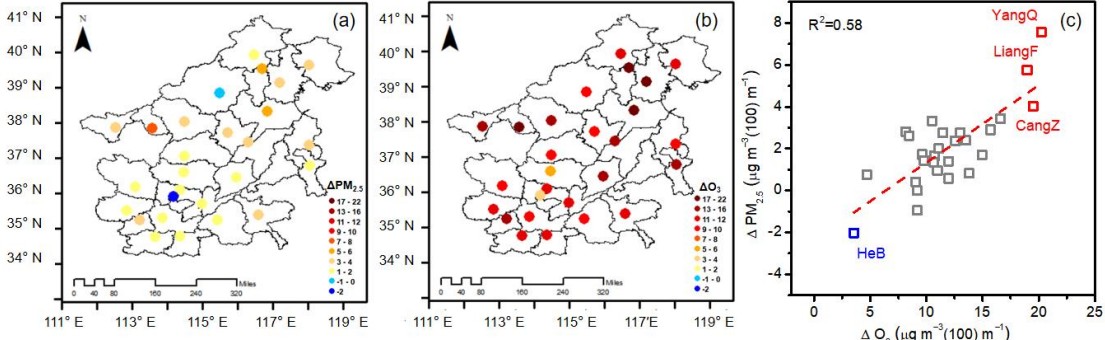


**Figure 13.** The spatial distribution of (a) $\Delta PM_{2.5}$ and (b) $\Delta O_3$. (c) The relationships between $\Delta PM_{2.5}$
and $\Delta O_3$ in the NCP during summertime. The corresponding correlation coefficients ($R^2$) was given at the
top of the panel.
**4 Conclusions**

531   Mixing layer height (MLH) was generally considered as a critical physical parameter in

atmospheric environmental evaluation. It is assumed that extended mixing layer may lead to the

dilution of air pollutants and thus tend to decrease surface concentrations. Several publications have

indeed reported such anti-correlations in cold seasons. However, the understanding of the interaction

between near surface $O_3$ and $PM_{2.5}$ (including its major components) along with the evolution of

mixing layer during warm seasons remained poor. Furthermore, previous observational studies were

mostly limited to a specific city. This paper is devoted to these topics by examining the response of

MDA8 $O_3$, $PM_{2.5}$, and its major components to the changes in mixing layer meteorology in the North

China Plain (NCP) during summertime. We showed that MDA8 $O_3$ initially increased and then

decreased with the growth of MLH. The maximum turning point of MLH was around 900–1800 m. As

for near-ground $PM_{2.5}$, similar non-linear change profile was found, with the maximum value of 31.65

$\mu g \ m^{-3}$ under medium MLH condition (900–1200 m), which was quite different from the results

conducted in cold seasons. Compared with winter, the occurrence of low MLH during summertime in

the NCP was mostly accompanied with cloudy or rainy conditions, which promoted wet deposition and

led to low concentrations of $PM_{2.5}$ at the ground level. Under medium MLH condition, strong chemical

productions of $SO_4^{2-}$ and OC occurred along with appropriate mixing layer meteorology, where RH

was around 50−70 %, and the availability of atmospheric oxidants (i.e., $O_3$) increased. The strong

chemical productions at medium MLH conditions may offset the diffusion effect on $PM_{2.5}$ induced by

mixing layer development, resulting in higher $PM_{2.5}$ levels. The chemical characteristics of $PM_{2.5}$

significantly changed along with the growth of MLH. The composited concentration of $NO_3^-$ was the

highest under low MLH condition, while the composited concentrations of $SO_4^{2-}$ and OC obviously

increased under medium MLH condition. Temperature was the key factor controlling the opposite

changes in $NO_3^-$ and $SO_4^{2-}$ concentrations in $PM_{2.5}$. We conclude that the MLH can be an indicator of

air pollutants in cold seasons, but the correlation between MLH and air pollutants, such as $O_3$ and

$PM_{2.5}$, should be treated with care in hot seasons. At least for the observation period in the NCP this

was not the case. Although several studies have examined the change characteristics of MLH and its

influence on ground-level $O_3$ and $PM_{2.5}$, it remains challenging to elucidate the mechanism underlying

the complex relationships. In this work, we did not quantify the sensitivity of $O_3$ and $PM_{2.5}$ to different

meteorological factors and chemical processes. To better understand the complex interactions between

MLH, air pollution, and chemical processing, a more detailed consideration the aids of explicit models

should be needed in the future. We also note that the present study was only confined to summertime

conditions (including two summer months) in the NCP, and the conclusions is likely to be different in
other seasons and regions. Thus, more extended observations in time and space should be needed in the
future.

**Data availability.** The data used in this paper can be provided upon request from the corresponding
author.

**Author contributions. J W and J G** conceived the study and designed the experiments. **J W, F C, X**
**Y,Y Y, L L and Y X** analyzed the data. **J W** prepared the manuscript and all the coauthors helped
improve the manuscript.

**Competing interests.** The authors declare that they have no conflict of interest.

**Acknowledgement.** We thank the platform of National Atmospheric Particulate
Chemical-Speciation-Network for making the $PM_{2.5}$ chemical composition data available.

**Financial support.** This work was supported by the National Natural Science Foundation of China (No.
42075182), the National research program for key issues in air pollution control (DQGG2021101) and
the Central Level, Scientific Research Institutes for Basic R&D Special Fund Business, China (No.
2022YSKY-26).

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
