# Peer review of "Summertime response of ozone and fine particulate"

_EGUsphere, 2023_

## Author Comment (AC1)

A point-by-point response to the reviews

Thank you for your valuable comments. The followings are our responses to your comments.

**Response to Reviewer #2**
**Comment 1:** The idea of this manuscript is clear. In this study, the composition of particulate matter and meteorological factors were deeply explored, which provided a good guidance for the collaborative treatment of ozone and particulate matter. The manuscript not only analyzed the relationship between $PM_{2.5}$ and $O_3$ and the boundary layer, but also focuses on the relationship between the components of $PM_{2.5}$ and the boundary layer, which is a very valuable part of this paper. However, it has much room for improvement. The analysis of this paper is rough, many things are unclear, and key parameters are still lacking verification. The author needs to supplement relevant information. Other suggestions for improvement are listed below. To sum up, It is strongly recommended to make significant revisions to the article, otherwise it cannot be accepted.

**Answer:** Thank you for your approval. According to your valuable comments, we have made significant revisions to our revised manuscript.

**Comment 2:** Lack of data profile description. For example, how many observatories are there, and what observation elements do each station have? The author should at least add a table that fully illustrates the data.

**Answer:** According to your valuable suggestions, the data profile description has been added in our revised manuscript (Page 6−7, line 111−143). Our field observations were conducted based on the existing ground-level observation stations (national control station, $PM_{2.5}$ component network, etc.) in the North China Plain (NCP), which covered two megacities (BeiJ and TianJ) and 26 surrounding cities. The spatial distribution of these 28 valid sites was shown in Figure 1. Hourly concentrations of ground-level $SO_2$, $NO_2$, $O_3$, $PM_{2.5}$ and its chemical compositions ($SO_4^{2-}$, $NO_3^-$, $NH_4^+$, and OC), and meteorological variables, including air temperature, relative humidity (RH), wind speed (WS) and direction (WD), and 24-h accumulated precipitation, at the sites were obtained from the platform of National Atmospheric Particulate Chemical-Speciation-Network. This network is established to improve the understanding of the heavy pollution formation mechanism in the North China Plain (NCP) and support the decision-making of local governments and state administration. Hourly $SO_2$, $NO_2$, $O_3$, $PM_{2.5}$ and its chemical compositions were recorded at the $PM_{2.5}$ component network, which was selected followed the Technical Regulation for Selection of Ambient Air Quality Monitoring Station published by the Ministry of Ecology and Environment of the People's Republic of China (HJ664−2013). The monitoring sites of $PM_{2.5}$ component network were mostly set up within the cities,

and can reflect the average pollution level of each city. Details for the near-ground observation stations of $PM_{2.5}$ component network were shown in Table R1. The meteorological variables were recorded in the national meteorological observation stations, and the information of each station can be obtained from the public website of China Meteorological Administration (http://data.cma.cn/data/cdcindex/cid/0b9164954813c573.html). It should be noted that the measurement sites of meteorological variables and air pollutants were not always consistent. To better analyze the meteorological conditions for $O_3$ and $PM_{2.5}$, only the station closed to the air quality monitoring station and representative of the city meteorological condition was selected in our work. The temporal resolution of air temperature, RH, WS and WD was 1-h. To avoid the influence of diurnal boundary layer cycles, in this article we focused on the relationships between daily mean air pollutants and meteorological factors. The daily mean meteorological factors, $PM_{2.5}$ and its major secondary components were calculated from the hourly data; daily $O_3$ concentration was characterized by the maximum daily 8 h average ozone (MDA8 $O_3$). Details for the near-ground observation species and the metrics were shown in Table R2. It's noted that when we rechecked the observation data, it's found that the calculated maximum daily 8 h average ozone (MDA8 $O_3$) in the original manuscript was incorrect, and the average of daily 8 h average ozone was misused. We have corrected the value of MDA8 $O_3$ in the whole article and carefully checked the accuracy of all observation data in the revised manuscript.

**Comment 3:** Figure 2 is confusing. My understanding is that the proportion of different values occurring at a certain time should sum to 100%. But the sum expressed in the figure must be more than 100%.

**Answer:** We have added the calculation method of the occurrence frequency (%) mentioned in Figure 2 in the methodology (Page 7, line 144−150). To better demonstrate the overall change characteristics of regional air pollution and meteorological conditions during the observation period, the occurrence frequency (%), which means the proportion of the number of cities at each air pollutant or methodology level, was calculated based on the following equation:

$$\text{Occurrence frequency}_X^{level} = \frac{N_X^{level}}{Total\ N_X} \times 100\% \tag{1}$$

where X means the air pollutant or methodology factors, $N_X^{level}$ represents the number of cities at each X level, $Total\ N_X$ represents the total number of cities. For example, as for the MLH condition, the MLHs were classified into 8 levels, and this ratio indicates the proportion of the number of cities at each MLH level to the total number of cites. As can be seen in Figure 2, on June 5, 2021, the proportion of the number of cities at MLH>2100 m was around 85 %, and significantly higher than other MLH conditions; on June 10, 2021, the MLH in all cities were lower than 1200 m, with the ratio at MLH<1200 as 100%.

**Comment 4:** L111 How much deviation are the two instruments for measuring the

composition of particulate matter? Were two different instruments used at different times? Please explain clearly.

**Answer:** MARGA (model ADI 2080) and AIM-IC (URG 9000D) have been widely utilized by government agencies and numerous research groups around the world for simultaneous measurements of water-soluble $PM_{2.5}$ constituents (Pang et al., 2021; Acharja et al., 2020; Liu et al., 2021; Liu et al., 2017; Vandenboer et al., 2014; Wang et al., 2022a; Ellis et al., 2011). They are similar in design and principle of operation. Detailed descriptions of the inlet design and the operating characteristics can be discovered in previous studies (Pang et al., 2020; Kong et al., 2018; Markovic et al., 2012; Gao et al., 2016). Previous works have shown that these two IC-based online instruments have shown good performance through instrument intercomparison studies or comparison to offline filters under clean to moderately polluted conditions (Markovic et al., 2012; Wu and Wang, 2007; Park et al., 2013; Rumsey et al., 2014). During the whole observation campaign, only one instrument (MARGA or AIM-IC) was used for ambient water-soluble $PM_{2.5}$ constituents monitoring in each station, and all instruments have been well operated to ensure the quality of observation data.

**Comment 5:** In view of the importance of MLH, the authors have not confirmed the calculation results. The authors use a very simple method to calculate the height of the mixed layer. It is suggested that the author make use of the meteorological profile or ERA5 reanalysis data to verify the reliability of the results.

**Answer:** Thank you for valuable comments. Even though the method for calculating MLH in this work seems simple, this methodology reflects the basic physical nature of the pollution mixing layer height. In recent years, many works have progressed in the atmospheric boundary layer characteristics, and analyzed the impacts of these parameter on air pollution. Planetary boundary layer (PBL), as one of the critical parameters to air quality modeling, has been well explored. However, PBL usually refers to the large-scale Ekman dynamic boundary layer (Haugen et al., 1971; Wang et al., 2014; Zhang et al., 2005). The way with which boundary layer describes the influences of air pollution is easily duplicated and confused (Niu et al., 2017). It is unreasonable to some extent, if the characteristic of the air pollution related to near-surface boundary layer is evaluated by using the concept of PBL. For air pollution measurement, one of selected functionalities of parameterization scheme for pollution mixing layer is to judge whether an air mass over a specific locality satisfies the "static and stable" attribute or not. Therefore, in this work, to express the basic physics for diagnosing meteorological conditions, we used the concept of pollution mixing layer height (MLH) proposed by (Wang et al., 2017), which was based on the classical synoptic theory according to the level of convective condensation layer, and the details of this method can be seen in previous work (Wang and Yang, 2000; Wang et al., 2017).

To be specific, we define the height close to the cloud base as the height of super-saturation layer (H_SSL), and the isoentropic atmospheric process meets the

level of convective condensation layer (LCL) in the super-saturation state, i.e., it is very close to the H_SSL. Iterative algorithm is used to work out the H_SSL (Wang and Yang, 2000):

$$\text{H\_SSL} \approx \text{LCL} = 6.11 \times 10^2 \times \left( \frac{0.622 + 0.622\frac{e_s}{p-e_s}}{0.622\frac{e_s}{p-e_s}} \right) \tag{2}$$

$$e_s = 6.22 \times \exp\frac{17.13(T-273.16)}{T-38} \tag{3}$$

where $e_s$ represents saturated water vapor pressure, T is temperature (K). Eq. (2) can be used to calculate the H_SSL, which is favorable for pollutant mixing and represented by (P). Below this height, the atmosphere gets supersaturated, causing the pollution mixing and wetting process in the low altitude to continue, so this height is also called the height of pollution mixing layer (MLH). Thus, MLH can be derived in the following expression:

$$\text{MLH} \approx \text{H\_SSL} \approx \text{LCL} = 6.11 \times 10^2 \times \left( \frac{0.622 + 0.622\frac{e_s}{p-e_s}}{0.622\frac{e_s}{p-e_s}} \right) \tag{4}$$

Several works have verified the reliability of the results based on this method. With this method, Wang et al. (2017) well characterized the features of mixing layer height in highly-sensitive areas of pollution in China ( 1−31 December 2015 for Beijing and the same period of 1−31 December 2015 for Guangzhou), and demonstrated the schematic diagram of 3-D model for low-level super-saturation layer and pollution mixing layer in the pollution hotspots in China, such as North China Plain (NCP), Yangtze River Delta (YRD), Pearl River Delta (PRD) and Si-Chuan Basin (SCB). Wang et al. (2022b) also used this method to explore the $PM_{2.5}$ and $O_3$ superposition-composite pollution event during spring 2020 in Beijing, China, and the hourly evolution of MLH, $O_3$, and $PM_{2.5}$ during the observation period were analyzed. The results can well depict the MLH diurnal cycle, which rises at daytime and decreases at night. In addition, Niu et al. (2017) has applied this method in Beijing, and the results showed that the pollution mixing layer can well present the change characteristics of haze pollution process. In this work, we further clarified the concept of MLH, and applied this method to investigate the impacts of MLH upon the change characteristics of ozone and fine particulate matter. The above discussion has been added in our revised manuscript (Page 7−8, line 152−190).

**Comment 6:** Almost all the graphs in Figure 3 show the scatter distribution, why does Figure c not has a scatter plot?

**Answer:** According to your valuable comments, we have adjusted the presentation form of Figure 3 to better show the scatter distribution in our revised manuscript (Figure R1).

**Comment 7:** Generally, when the boundary layer rises, the wind speed will increase, especially when the boundary layer exceeds 1500 m, but this phenomenon is not

shown in this study. In addition, in general, when precipitation occurs, strong convection occurs, and the height of the mixing layer will suddenly rise, which is different from the author's study. Therefore, it is highly recommended that the authors validate the results of mixing layer height.

**Answer:** Thanks for your valuable comments. As we known, there are differences between MLH and PBLH (Height of planetary Boundary layer). These phenomena were generally summarized based on PBLH in individual cases. Besides, these phenomena can not fit each case, and there are still exceptions. According to your suggestions, we added the change characteristics of wind speed (WS) along with the increase of MLH (Figure R2 and R3). Actually, we can see apparent increase of WS when MLH in the range of 0−300 m which was probably due to precipitation events. The increase of WS when MLH exceeds 1500 have also been observed, but the increment was not so obvious. Previous works by Liu and Liang (2010) and Li et al. (2020) have found that the severe convective weather generally decreases PBLH, and the precipitation was highly negatively correlated with PBLH, which was consistent with the results found in our work. The rainfall events may produce clouds, then reduce surface solar and thermal heating, thus suppressing the PBLH. The reliability of the results based on this MLH calculation method has been verified by many works. The calculated MLHs showed strong diurnal patterns, and can well depict the change characteristics of haze pollution processes (Niu et al., 2017; Wang et al., 2022b) .

**Comment 8:** It is more appropriate to reflect the atmospheric oxidation capacity with the change of Ox.

**Answer:** The aim of Part 3.4 was to explore the superposition-composite effects and the interaction between $PM_{2.5}$ and $O_3$ along with the evolution of mixing layer. According to your valuable suggestions, the title of Part 3.4 has been replaced by "Superposition-composite effects of $PM_{2.5}$ and $O_3$ with the evolution of mixing layer". (Page 19, line 392)

**Comment 9:** Figure 9 can be represented as a time-by-time coloring plot, as the mean may mask the characteristics of high-altitude transport. And ozone profile results are extremely abnormal, with very little ozone decline with altitude. Ozone radar has great shortcomings in the summer when there are clouds and high humidity. Due to the extremely fast attenuation of shortwave radiation, radar echo signal will decay rapidly under the influence of water vapor, thus affecting the observation results. It is strongly suggested that delete this content.

**Answer:** According to your valuable suggestions, this content has been deleted in our revised manuscript. To reveal the impact of regional transport, we have presented the change characteristics of WS and WD during the typical $PM_{2.5}$-$O_3$ co-polluted episode.

**Comment 10:** The authors need to find a case to fully present the relationship between the mixing layer and pollutants, and use the hourly concentration to illustrate the response of PM, its components and ozone to the mixing layer.

**Answer:** According to the National Ambient Air Quality Standard of China (GB3095-2012), $O_3$ ($PM_{2.5}$) concentration exceeds the national air quality standard if the MDA8 $O_3$ (daily $PM_{2.5}$ average) concentration higher than 160 $\mu g\,m^{-3}$ (75 $\mu g\,m^{-3}$). The daily $PM_{2.5}$ averages in "2+26" cities can meet the Level II national ambient air quality standard (75 $\mu g\,m^{-3}$), while exceeding the level I standard of 35 $\mu g\,m^{-3}$. Here, we defined a $O_3$–$PM_{2.5}$ co-polluted episode as a set of continuous days (longer than 4 days) with MDA8 $O_3$ and daily mean $PM_{2.5}$ in more than 10 % NCP cities exceeding 160 $\mu g\,m^{-3}$ and 35 $\mu g\,m^{-3}$, respectively. According to this criterion, three typical $O_3$–$PM_{2.5}$ co-polluted episodes were selected: June 4–14 (Episode I), June 18−29 (Episode II), and July 2–11 (Episode III), 2021, and these three episodes have been marked in Figure 2.

[revised manuscript text omitted]

To explore the relevance of hourly $O_3$, $PM_{2.5}$, its components and MLH, we have taken PuY and HeZ as examples. Figure R7 plotted the day-to-day variations along with the diurnal variations of $O_3$, $PM_{2.5}$, its components and MLH in PuY and HeZ during Episode II (June 18−29, 2021). The results showed that there were large diurnal as well as day-to-day variability in the $O_3$ and $PM_{2.5}$ levels. The diurnal variations of MLH were clearly visible (Figure R8), with the rise in MLH during the daytime and the decrease in MLH at night. The concentration of $PM_{2.5}$ increased with the decrease of MLH at night, but the concentration of $O_3$ increased with the rise of MLH at daytime. Interestingly, we observed noontime soar of $SO_4^{2-}$ and OC concentrations in PuY, and the values of SOR kept stable or even increased at noon. Besides, it's noted that daily $O_3$ and $PM_{2.5}$ both gradually accumulated with the increase of MLH during June 18−21 and 26−28, which can be attributed to the $O_3$ and $PM_{2.5}$ superposition composite effects. The decrease in $PM_{2.5}$ at daytime with the rise of MLH can be partly offset by an increment in secondary pollutants formation derived from $O_3$ growth. Then with the decrease of MLH at night, the concentration of the original existing $PM_{2.5}$ increased due to unfavorable diffusion. In general, the conclusions in this work was only suitable to the day-to-day relationship between air pollutants and MLH. The hourly relationships were more complicated and need more further analysis. According to your valuable comments, we will further explore the hourly relationship in the NCP in our follow-up studies. The above discussion has been added in our revised manuscript (Page 19−24, line 393−476).

**Comment 11**: As discussed in Figure 10, it is suggested that the concentration classification of MDA8 ozone can be further refined to make the change of pollutants more obvious.

**Answer:** According to your valuable suggestions, we have further refined the classification of MAD8 $O_3$ in Figure R9 ($< 140$ µg m$^{-3}$, 140–160 µg m$^{-3}$, 160–180 µg m$^{-3}$, 180–200 µg m$^{-3}$, $> 200$ µg m$^{-3}$). The concentrations of PM$_{2.5}$ and its major components increased synchronously with elevated MDA8 $O_3$ concentration, especially when MDA8 $O_3$ increased from $< 140$ to 180–200 µg m$^{-3}$. With elevated MDA8 $O_3$ concentration, SOR and NOR both slightly increased, and reached the maximum when MDA8 $O_3$ at around 160–200 µg m$^{-3}$, which indicated the strong secondary formation of $SO_4^{2-}$ and $NO_3^-$ promoted by high $O_3$ concentration. When MDA8 $O_3$ increased from 180–200 to $> 200$ µg m$^{-3}$, the concentrations of $NO_3^-$, $NH_4^+$, and $SO_4^{2-}$ kept stable or started to decrease, and the values of SOR and NOR decreased synchronously. During this stage, the high $O_3$ concentration often accompanied with dry and hot meteorological conditions, which was not beneficial to the aqueous chemical production and was conducive to the partitioning of nitrate to the gas phase. (Page 25, line 485−498)

**Comment 12**: In Figure 11, it is suggested to use red for positive correlation and blue for negative correlation, with the same color scale for ozone and particulate matter. Looking at the figure, the slope of particulate matter and MLH is very low, indicating that MLH has little effect on particulate matter. Please discuss the linear p-value and R2, if there is no significant correlation, it is meaningless to discuss the rate of change alone.

**Answer:** Thanks to your valuable comments. We are sorry to make the mistakes about the units of $\Delta$PM$_{2.5}$ in Figure 11. The unit of in the original manuscript was µg m$^{-3}$ m$^{-1}$, and we multiplied the values by 100 m in our revised manuscript (µg m$^{-3}$ (100) m$^{-1}$). According to your valuable suggestions, we have changed and unified the color scale of for ozone and particulate matter, with red for positive correlation and blue for negative correlation in Figure R10.

**Comment 13**:Technical comments:
1) L221 fund? found?
2) Figure 5a and 5b It is recommended that the color of the value be uniform from small to large.

**Answer:** Thanks to your valuable comments. Line 211 has been rephrased (Page 14) and the color of the values in Figure 5a and 5b have been uniformed from small to large in our revised manuscript (Figure R11 and R12).

**Table R1.** List of observation stations and locations.

| No. | Site | Abbreviation | Station | longitude (°E) | latitude (°N) |
|---|---|---|---|---|---|
| 1 | BeiJing | BeiJ | China National Environmental Monitoring Centre | 116.41 | 40.04 |
| 2 | Tianjin | TianJ | Zhongshan North Road Station | 117.21 | 39.17 |
| 3 | Shijiazhuang | SJZ | Northwest Shuiyuan Station | 114.49 | 38.13 |
| 4 | Langfang | LangF | Langfang Hebei University of Technology Station | 116.70 | 39.55 |
| 5 | Baoding | BaoD | Yangguang North Street Station | 115.48 | 38.93 |
| 6 | Tangshan | TangS | Xiaoshan Station | 118.19 | 39.62 |
| 7 | Handan | HanD | Guangming South Street Station | 114.50 | 36.57 |
| 8 | Hengshui | HengS | Hengshui Ecology and Environment Bureau Station | 115.68 | 37.73 |
| 9 | Xingtai | XingT | Quanbei Street Station | 114.53 | 37.09 |
| 10 | Cangzhou | CangZ | Cangzhou Technical College Station | 116.82 | 38.28 |
| 11 | Taiyuan | TaiY | Taiyuan Jinyuan Station | 112.48 | 37.71 |
| 12 | Yangquan | YangQ | Nanzhuang Road Station | 113.59 | 37.85 |
| 13 | Changzhi | ChangZ | Changzhi Ecology and Environment Bureau Station | 113.11 | 36.20 |
| 14 | Jincheng | JinC | Jincheng Ecology and Environment Bureau Station | 112.86 | 35.49 |
| 15 | Jinan | JiNan | Jinan Environmental Monitoring Station | 117.06 | 36.66 |
| 16 | Zibo | ZiB | Beijing Road station | 117.91 | 36.84 |
| 17 | Jining | JiNing | Jinyu Road Station | 116.63 | 35.43 |
| 18 | Dezhou | DeZ | Baima Lake Station | 115.83 | 36.95 |
| 19 | Liaocheng | LiaoC | Liaocheng monitoring center Station | 115.98 | 36.50 |
| 20 | Binzhou | BinZ | Binzhou Ecology and Environment Bureau Station | 118.01 | 37.38 |
| 21 | Heze | HeZ | Heze Quality Supervision | 115.53 | 35.21 |

| | | | Bureau Station | | |
|---|---|---|---|---|---|
| 22 | Zhengzhou | ZhengZ | Zhengzhou Forty-seven Middle School Station | 113.74 | 34.77 |
| 23 | Kaifeng | KaiF | Jinming West Street Station | 114.30 | 34.80 |
| 24 | Anyang | AnY | Anyang Ecology and Environment Bureau Station | 114.40 | 36.09 |
| 25 | Hebi | HeB | Hebi Ecology and Environment Bureau Station | 114.29 | 35.72 |
| 26 | Xinxiang | XinX | Xinxiang Ecology and Environment Bureau Station | 113.92 | 35.30 |
| 27 | Jiaozuo | JiaoZ | Fengshou Middle Road Station | 113.28 | 35.21 |
| 28 | Puyang | PuY | Jinti Road Station | 115.04 | 35.76 |

**Table R2.** List of observation species and metrics.

| Species | Unit | Temporal resolution | Metrics |
|---|---|---|---|
| Gaseous pollutants | | | |
| $O_3$ | $\mu g\ m^{-3}$ | 1 h | Maximum daily 8 h average |
| $SO_2$ | $\mu g\ m^{-3}$ | 1 h | Daily average |
| $NO_2$ | $\mu g\ m^{-3}$ | 1 h | Daily average |
| $PM_{2.5}$ and its major components | | | |
| $PM_{2.5}$ | $\mu g\ m^{-3}$ | 1 h | Daily average |
| $SO_4^{2-}$/ $NO_3^-$/ $NH_4^+$ | $\mu g\ m^{-3}$ | 1 h | Daily average |
| OC | $\mu g\ m^{-3}$ | 1 h | Daily average |
| Meteorological variables | | | |
| Temperature (T) | °C | 1 h | Daily average |
| Relative Humidity (RH) | % | 1 h | Daily average |
| Wind speed (WS) | $m\ s^{-1}$ | 1 h | Daily average |
| Wind direction (WD) | ° | 1 h | Daily average |
| 24-h precipitation | mm | 24 h | 24-h accumulated |

[Figure]

**Figure R1.** The variation characteristics of (a) $SO_4^{2-}$, (b) $NO_3^-$, (c) $NH_4^+$, and (d) OC in different MLH conditions. Box plots show the inter quartile range (the distance between the bottom and the top of the box), median (the band inside the box), and 95 % confidence interval (whiskers above and below the box) of the data.

[Figure]

**Figure R2.** The change characteristics of WS under different MLH conditions.

[Figure]

**Figure R3.** (a) The overall WS and WD condition during the observation campaign, (b) the change characteristics of WS and WD under different MLH levels. S: south; N: north; E: east; W: west.

[Figure]

**Figure R4.** The spatial distribution of (a) MLH, (b) MDA8 $O_3$, (c) $PM_{2.5}$, (d) the dominant $PM_{2.5}$ chemical component (N: $NO_3^-$ dominant, NS: $NO_3^-$ and $SO_4^{2-}$ dominant, NO: $NO_3^-$ and OM dominant, S: $SO_4^{2-}$ dominant, SN: $SO_4^{2-}$ and $NO_3^-$ dominant, SO: $SO_4^{2-}$ and OM dominant, O: OM dominant, ON: OM and $NO_3^-$ dominant, OS: OM and $SO_4^{2-}$ dominant), (e) T, and (f) RH, (g) the overall change characteristics of WS and WD in the NCP from June 18 to 23, 2021.

[Figure]

**Figure R5.** The spatial distribution of (a) MLH, (b) MDA8 O$_3$, (c) PM$_{2.5}$, (d) the dominant PM$_{2.5}$ chemical component (N: NO$_3^-$ dominant, NS: NO$_3^-$ and SO$_4^{2-}$ dominant, NO: NO$_3^-$ and OM dominant, S: SO$_4^{2-}$ dominant, SN: SO$_4^{2-}$ and NO$_3^-$ dominant, SO: SO$_4^{2-}$ and OM dominant, O: OM dominant, ON: OM and NO$_3^-$ dominant, OS: OM and SO$_4^{2-}$ dominant), (e) T, and (f) RH, (g) the overall change characteristics of WS and WD in the NCP from June 24 to 29, 2021.

[Figure]

**Figure R6.** The distribution characteristics of (a) NOR and SOR, and (b) the mass fractions of major $PM_{2.5}$ components, MLH, RH, and T under $O_3$–$PM_{2.5}$ CPD, $O_3$ PD, $PM_{2.5}$ PD, and NPD conditions from June 24 to 29, 2021.

[Figure]

**Figure R7.** The hourly evolution of $O_3$, $PM_{2.5}$, its components and MLH in HeZ and PuY during June 18−29, 2021.

[Figure]

**Figure R8.** The diurnal variation of MLH, SOR, NOR, O₃, PM₂.₅, and its components in HeZ and PuY during June 18−29, 2021.

[Figure]

**Figure R9.** Box plots showing the statistics of (a) $PM_{2.5}$, (b) NOR and SOR, (c) $SO_4^{2-}$, (d) $NO_3^-$, (e) $NH_4^+$, and (f) OC for different MDA8 $O_3$ conditions ($< 140$ µg m$^{-3}$, 140–160 µg m$^{-3}$, 160–180 µg m$^{-3}$, 180–200 µg m$^{-3}$, $> 200$ µg m$^{-3}$). The distance between the bottom and the top of the box reflects the inter quartile range; the line and square in between are the median and mean values, respectively. The whiskers above and below the box refer the 95 % confidence interval of the data. Note that rainy days were excluded.

[Figure]

**Figure R10.** The spatial distribution of (a) $\Delta PM_{2.5}$ and (b) $\Delta O_3$. (c) The relationships between $\Delta PM_{2.5}$ and $\Delta O_3$ in the NCP during summertime. The corresponding correlation coefficients ($R^2$) was given at the top of the panel.

[Figure]

**Figure R11.** The distribution characteristics of the MDA8 O$_3$ concentrations with the evolution of MLH under different (a) temperature, (b) RH, (c) precipitation, (d) WS, and (e) WD conditions.

[Figure]

**Figure R12.** The distribution characteristics of the PM$_{2.5}$ concentrations with the evolution of MLH under different (a) temperature, (b) RH, (c) precipitation, (d) WS, and (e) WD conditions.

---

## Author Comment (AC2)

A point-by-point response to the reviews

Thank you for your valuable comments. The followings are our responses to your comments.

**Response to Reviewer #1**

**Comment 1:** The article contains highly relevant results for the understanding of ozone and secondary PM episodes. This is especially important in areas where the WHO guidelines are widely exceeded. The relationship between the MLH and the measured concentrations can help diagnose more precisely the underlying causes of these high pollution episodes during the summer. Consequently, it is relevant for proper air quality control strategies.

While the results are important and support the authors' conclusions, there seems to be a lack of depth in their analysis. Additionally, there is a noticeable absence of indication regarding potential limitations and uncertainties in this study.

**Answer:** Thank you for your approval. According to your valuable comments, we have made significant revisions to our revised manuscript. The purpose of this work was to gain new insights into the underlying causes of the summertime $O_3$–$PM_{2.5}$ coordinated pollution through exploring the response of ozone and fine particulate matter to mixing layer meteorology over the North China Plain from June 1 to July 31, 2021. However, there remained some limitations and uncertainties in this work. First, the present study was only confined to summertime conditions (including two summer months) in the NCP, and the conclusions was likely to be different in other seasons and regions. Thus, more extended observations in time and space should be needed in the future. Second, to avoid the influence of diurnal boundary layer cycles, in this article we focused on the relationships between daily mean air pollutants and meteorological factors. We have also taken PuY and HeZ as examples to explore the relevance of hourly $O_3$, $PM_{2.5}$, its components and MLH. The results showed large diurnal as well as day-to-day variability in the $O_3$ and $PM_{2.5}$ levels. The decrease in $PM_{2.5}$ at daytime with the rise of MLH can be partly offset by an increment in secondary pollutants formation derived from $O_3$ growth. In general, the conclusions in this work was only suitable to the day-to-day relationship between air pollutants and MLH. The hourly relationships were much more complicated and need more further analysis. Lastly, a weakness of this study is that we did not quantify the sensitivity of $O_3$ and $PM_{2.5}$ to different meteorological factors and chemical processes, thus a more detailed consideration with the aid of modeling would be needed in the future. Such an exploration requires efforts going beyond the current project, and it is therefore not pursued. The above discussion has been added in our revised manuscript. (Page 21, line 449−451; page 26, line 521−525; page 28, line 558−564)

**Comment 2:** The analyzed results are limited to only two summer months, which may not be easily extrapolated to other summers. It is essential to address why the

three mentioned episodes are relevant and whether the selected two months are representative of typical patterns observed throughout the years. Were there any meteorological anomalies during this period? These aspects should clear throughout the entire manuscript to provide a more comprehensive understanding of the study's scope and implications.

**Answer:** Thank you for your valuable comments. We considered the months of June and July can well represent the typical characteristics of $O_3$–$PM_{2.5}$ coordinated pollution during warm seasons in the North China Plain (NCP). According to the hourly concentrations of $PM_{2.5}$ and MDA8 $O_3$ in China over the years of 2013–2020, the observed numbers of $O_3$ polluted days (MDA8 $O_3$ concentration $> 160\ \mu g\ m^{-3}$ and $PM_{2.5} < 75\ \mu g\ m^{-3}$), $PM_{2.5}$ polluted days (MDA8 $O_3$ concentration $< 160\ \mu g\ m^{-3}$ and $PM_{2.5} > 75\ \mu g\ m^{-3}$), and $O_3$–$PM_{2.5}$ co-polluted days (MDA8 $O_3$ concentration $> 160\ \mu g\ m^{-3}$ and $PM_{2.5} > 75\ \mu g\ m^{-3}$) were highest in the North China Plain (NCP). Besides, $O_3$–$PM_{2.5}$ co-polluted days and $O_3$ polluted days were generally occurred in June and July (Dai et al., 2023). According to the National Ambient Air Quality Standard of China (GB3095-2012), the daily $PM_{2.5}$ averages in "2+26" cities can meet the Level II national ambient air quality standard ($75\ \mu g\ m^{-3}$), while exceeding the level I standard of $35\ \mu g\ m^{-3}$. Here, we defined a $O_3$–$PM_{2.5}$ co-polluted episode as a set of continuous days (longer than 4 days) with MDA8 $O_3$ and daily mean $PM_{2.5}$ in more than 10 % NCP cities exceeding $160\ \mu g\ m^{-3}$ and $35\ \mu g\ m^{-3}$, respectively. According to this criterion, three typical $O_3$–$PM_{2.5}$ co-polluted episodes were selected: June 4–14 (Episode I), June 18–29 (Episode II), and July 2–11 (Episode III), 2021. These three episodes have been marked in Figure 2 in our revised manuscript. Comparing with the same observation period in 2020, the temperature was slightly higher and more precipitation evens happened during the summertime in 2021 (National Bulletin of Atmospheric Environment, 2021; http://www.nmc.cn/publish/environment/National-Bulletin-atmospheric-environment. htm). Our work can provide an overall diagnosis of the response of ozone and fine particulate matter to mixing layer meteorology during summertime, especially in warm and humid seasons, in the NCP.

**Comment 3:** The acronyms OC and OM are mixed.

**Answer:** Thank you for your valuable comments. The mistakes have been corrected in our revised manuscript. The concentration of OM can be obtained by multiplying the OC concentration by a factor of 1.6 (Li et al., 2021). When discussing the change characteristics of mass concentration, we use acronyms OC, while when analyzing the mass fraction in $PM_{2.5}$, the acronyms OM was used.

**Comment 4:** In the graphical abstract, it would be more convenient to increase the font size of the particle compounds. Regarding the arrow for ozone at the MLH between 900 and 1200, what does it mean? Is ozone not increasing, or is it at its maximum at that MLH? It would be clearer to have a separate arrow for $O_3$ and

$PM_{2.5}$.

**Answer:** The arrow for ozone at the MLH between 900 m and 1200 m means the significant increase of $O_3$. According to your suggestions, we have adjusted the font size of the particle compounds and separated the arrows for $O_3$ and $PM_{2.5}$ in the graphical abstract in our revised manuscript (Figure R1).

**Comment 5:** I feel that the statement on line 80 needs further elaboration. It goes from negative to positive, but why? Could it be due to them being more of a secondary type rather than primary, or have there been changes in emissions?

**Answer:** As indicated in previous studies (Chu et al., 2020), the correlation between $PM_{2.5}$ and $O_3$ tended to change from negative to positive as air quality improved (when the $PM_{2.5}$ concentration lower than 50 $\mu g\,m^{-3}$). One speculative reason for this phenomenon is that $PM_{2.5}$ does not reduce actinic flux and $HO_2$ radical significantly when the $PM_{2.5}$ concentration was low in summer. On the other hand, $PM_{2.5}$ and $O_3$ tend to be positively correlated possibly due to their common precursors, such as VOCs and NOx, and their simultaneous generation in photochemical reactions. In addition, the generation of $O_3$ would also catalyze the generation of the secondary $PM_{2.5}$, and both showed a trend of growing together (Wu et al., 2022). (Page 4, line 79–88)

**Comment 6:** Line 95, are those months the most "problematic"?

**Answer:** Yes, according to the hourly concentrations of $PM_{2.5}$ and MDA8 $O_3$ in China over the years of 2013–2020, the observed numbers of $O_3$ polluted days, $PM_{2.5}$ polluted days, and $O_3$–$PM_{2.5}$ co-polluted days were the highest in June and July in the NCP. Therefore, we considered June and July can represent the typical characteristics of $O_3$–$PM_{2.5}$ co-polluted days during warm seasons in the NCP.

**Comment 7:** In the map of Figure 1, it is advisable to specify the exact locations of the measurements. Are they conducted within cities? What environments have been selected, and what criteria were used for their selection? Is there any measurement that could be heavily influenced by local emissions? Additionally, another legend could indicate the locations of the soundings (vertical profiles) shown in Figure 9.
Line 106, the meteorological variables are measured at the same measurement site, right?
What is the temporal resolution of the meteorological measurements? Have they been averaged to obtain daily averages? It is not clear.
Line 117, "other pollutants" is too vague. What does it refer to?

**Answer:** According to your valuable suggestions, the data profile description has been added in our revised manuscript (Page 6−7, line 111−143). Our field observations were conducted based on the existing ground-level observation stations

(national control station, PM$_{2.5}$ component network, etc.) in the North China Plain (NCP), which covered two megacities (BeiJ and TianJ) and 26 surrounding cities. The spatial distribution of these 28 valid sites was shown in Figure 1. Hourly concentrations of ground-level SO$_2$, NO$_2$, O$_3$, PM$_{2.5}$ and its chemical compositions (SO$_4^{2-}$, NO$_3^-$, NH$_4^+$, and OC), and meteorological variables, including air temperature (T), relative humidity (RH), wind speed (WS) and direction (WD), and 24-h accumulated precipitation, at the sites were obtained from the platform of National Atmospheric Particulate Chemical-Speciation-Network. This network is established to improve the understanding of the heavy pollution formation mechanism in the North China Plain (NCP) and support the decision-making of local governments and state administration. Hourly SO$_2$, NO$_2$, O$_3$, PM$_{2.5}$ and its chemical compositions were recorded at the PM$_{2.5}$ component network, which was selected followed the Technical Regulation for Selection of Ambient Air Quality Monitoring Station published by the Ministry of Ecology and Environment of the People's Republic of China (HJ664−2013). The monitoring sites of PM$_{2.5}$ component network were mostly set up within the cities, and can reflect the average pollution level of each city. Details for the near-ground observation stations of PM$_{2.5}$ component network were shown in Table R1. The meteorological variables were recorded in the national meteorological observation stations, and the information of each station can be obtained from the public website of China Meteorological Administration (http://data.cma.cn/data/cdcindex/cid/0b9164954813c573.html). It should be noted that the measurement sites of meteorological variables and air pollutants were not always consistent. To better analyze the meteorological conditions for O$_3$ and PM$_{2.5}$, only the station closed to the air quality monitoring station and representative of the city meteorological condition was selected in our work. The temporal resolution of air temperature, RH, WS and WD was 1-h. To avoid the influence of diurnal boundary layer cycles, in this article we focused on the relationships between daily mean air pollutants and meteorological factors. The daily mean meteorological factors, PM$_{2.5}$ and its major secondary components were calculated from the hourly data; daily O$_3$ concentration was characterized by the maximum daily 8 h average ozone (MDA8 O$_3$). Details for the near-ground observation species and the metrics were shown in Table R2. It's noted that when we rechecked the observation data, it's found that the calculated maximum daily 8 h average ozone (MDA8 O$_3$) in the original manuscript was incorrect, and the average of daily 8 h average ozone was misused. We have replaced the value of MDA8 O$_3$ in the whole article and carefully checked the accuracy of all observation data in the revised manuscript.

"Other pollutants" refers to NO$_2$ and SO$_2$, and we have made it clear in our revised manuscript. (Page 6, line 132)

According to the valuable suggestion of Reviewer #2, the ozone radar has great shortcomings in the summer when there are clouds and high humidity, and the data may send wrong message and misled readers. Therefore, we deleted the soundings part in our revised manuscript.

**Comment 8:** Point 2.2. The methodology for calculating the MLH seems somewhat

simplistic. While estimating MLH can be complex, it would be advisable to compare the results with sounding data (already shown) or reanalysis data to determine their coherence with the calculated values.

**Answer:** Thank you for valuable comments. Even though the method for calculating MLH in this work seems simple, this methodology reflects the basic physical nature of the pollution mixing layer height. In recent years, many works have progressed in the atmospheric boundary layer characteristics, and analyzed the impacts of these parameter on air pollution. Planetary boundary layer (PBL), as one of the critical parameters to air quality modeling, has been well explored. However, PBL usually refers to the large-scale Ekman dynamic boundary layer (Haugen et al., 1971; Wang et al., 2014; Zhang et al., 2005). The way with which boundary layer describes the influences of air pollution is easily duplicated and confused (Niu et al., 2017). It is unreasonable to some extent, if the characteristic of the air pollution related to near-surface boundary layer is evaluated by using the concept of PBL. For air pollution measurement, one of selected functionalities of parameterization scheme for pollution mixing layer is to judge whether an air mass over a specific locality satisfies the "static and stable" attribute or not. Therefore, in this work, to express the basic physics for diagnosing meteorological conditions, we used the concept of pollution mixing layer height (MLH) proposed by Wang et al. (2017), which was based on the classical synoptic theory according to the level of convective condensation layer, and the details of this method can be seen in previous work (Wang and Yang, 2000; Wang et al., 2017).

To be specific, we defined the height close to the cloud base as the height of super-saturation layer (H_SSL), and the isoentropic atmospheric process meets the level of convective condensation layer (LCL) in the super-saturation state, i.e., it is very close to the H_SSL. Iterative algorithm is used to work out the H_SSL (Wang and Yang, 2000):

$$H\_SSL \approx LCL = 6.11 \times 10^2 \times \left( \frac{0.622 + 0.622 \frac{e_s}{p - e_s}}{0.622 \frac{e_s}{p - e_s}} \right) \tag{1}$$

$$e_s = 6.22 \times \exp \frac{17.13(T - 273.16)}{T - 38} \tag{2}$$

where $e_s$ represents saturated water vapor pressure, T is temperature (K). Eq. (1) can be used to calculate the H_SSL which is favorable for pollutant mixing and represented by (P). Below this height, the atmosphere gets supersaturated, causing the pollution mixing and wetting process in the low altitude to continue, so this height is also called the height of pollution mixing layer (MLH). Thus, MLH can be derived in the following expression:

$$MLH \approx H\_SSL \approx LCL = 6.11 \times 10^2 \times \left( \frac{0.622 + 0.622 \frac{e_s}{p - e_s}}{0.622 \frac{e_s}{p - e_s}} \right) \tag{3}$$

Several works have verified the reliability of the results based on this method. With this method, Wang et al. (2017) well characterized the features of mixing layer

height in highly-sensitive areas of pollution in China ( 1–31 December 2015 for Beijing and the same period of 1–31 December 2015 for Guangzhou), and demonstrated the schematic diagram of 3-D model for low-level super-saturation layer and pollution mixing layer in the pollution hotspots in China, such as North China Plain (NCP), Yangtze River Delta (YRD), Pearl River Delta (PRD) and Si-Chuan Basin (SCB). Wang et al. (2022) also used this method to explore the $PM_{2.5}$ and $O_3$ superposition-composite pollution event during spring 2020 in Beijing, China, and the hourly evolution of MLH, $O_3$, and $PM_{2.5}$ during the observation period were analyzed. The results can well depict the MLH diurnal cycle, which rises at daytime and decreases at night. In addition, Niu et al. (2017) has applied this method in Beijing, and the results showed that the pollution mixing layer can well present the change characteristics of haze pollution process. In this work, we further clarified the concept of MLH, and applied this method to investigate the impacts of MLH upon the change characteristics of ozone and fine particulate matter. The above discussion has been added in our revised manuscript (Page 7–8, line 152–190).

**Comment 9:** The % ratio of occurrence shown in Figure 2 should be mentioned in the methodology. How has it been calculated? The WS (Wind Speed) is barely mentioned, and I miss the Wind Direction. Although the authors discard advection, how can authors be certain that it is not always due to the same emission source or the same synoptic pattern?

**Answer:** Thank you for valuable comments. We have added the calculation method of the occurrence frequency (%) mentioned in Figure 2 in the methodology (Page 7, line 144–150). To better demonstrate the overall change characteristics of regional air pollution and meteorological conditions, the proportion of the number of cities at each air pollutant or methodology level was calculated based on the following equation:

$$\text{0ccurrence frequency}_X^{\text{level}} = \frac{N_X^{\text{level}}}{\text{Total } N_X} \times 100\% \tag{4}$$

where X means the air pollutants or methodology factors, $N_X^{\text{level}}$ represents the number of cities at each X level, $\text{Total } N_X$ represents the total number of cities. For example, as for the MLH condition, the MLHs were classified into 8 levels, and this ratio indicates the proportion of the number of cities at each MLH level to the total number of cites. As can be seen in Figure 2, on June 5, 2021, the proportion of the number of cities at MLH>2100 m was around 85 %, and significantly higher than other MLH conditions; on June 10, 2021, the MLHs in all cities were lower than 1200 m, with the ratio at MLH<1200 as 100%.

According to your suggestions, we also presented the results of wind speed (WS) and wind direction (WD) to reveal the impact of wind on the variation trends of air pollutions in our revised manuscript (Figure R2– R6). As shown in Figure R2, during the whole campaign, the NCP was dominated by winds from northeast and south (45–225 °). Because more than 75 % WD were in the rang of 45–225 °, the WD was classified into 4 categories: 45–90, 90–135, 135–180, and 180–225 °. To further investigate the impact of wind on MDA8 $O_3$ and $PM_{2.5}$ concentrations, we also

demonstrated the statistics on the concentration distributions of MDA8 $O_3$, $PM_{2.5}$ and its dominant components with the increase of MLH under different WS and WD conditions in Figure R4–R6. In general, WS could affect the diffusion of air pollutants. During the observation period, WS was mostly less than 3 m s$^{-1}$, and the concentrations of air pollutants were comparatively higher at low wind speeds. As shown in Figure R3, at low MLH conditions (MLH<300 m), a northeasterly wind prevailed near the ground, and the WS was generally higher than other conditions. The concentration of MDA8 $O_3$ was low during this period. With the increase of MLH, the WD gradually changed from northeast (MLH=300–600 m) to southeast (MLH=600–900 m) and south (MLH=900–1200 m). The North China Plain (NCP) is surrounded to the west by the Taihang Mountains, to the north by the Yan Mountains, and to the east by the Bohai Sea (Figure 1). The southerly wind can transport the gaseous pollutant or $O_3$ from the southern part of the plain area to the northern part, and the Taihang mountains may block pollutant transport, leading to the accumulation of pollutants along the foot of the Taihang Mountains. It's noted that the concentration of MDA8 $O_3$ was higher when the plain dominated by southerlies (180–225 °) (Figure R4). When MLH higher than 2100 m, NCP was governed by northwest winds, which preferred the outward transport and diffusion of pollution, leading to the decrease of MDA8 $O_3$. Comparing with $O_3$, the impact of WD along with the increase of MLH seems different for $PM_{2.5}$ and its dominant components. When MLH in the range of 600–1200 m, the NCP was dominated by southeast or south winds. However, when southeast or south wind prevailed, the corresponding $PM_{2.5}$ and its dominant components concentrations were comparable or even lower than other WD situations (Figure R5–R6). This indicated that regional transport was not the dominant factor leading to the elevation of $PM_{2.5}$ and its aerosol species along with the evolution of mixing layer (MLH < 1200 m).

**Comment 10:** It would be interesting to mark some of the issues mentioned in the text in Figure 2 to assist the reader.

**Answer:** We have marked these three typical episodes: June 4–14 (Episode I), June 18–30 (Episode II), and July 2–11 (Episode III), 2021) in Figure 2 and Figure S1 in our revised manuscript.

**Comment 11:** In the sentence starting at line 153, it lacks clarification on "how."
Although it is mentioned later.
The sentence in 161 is not clear.

**Answer:** Thank you for valuable comments. The content in Section 3.1 has been reorganized and revised in our manuscript (Page 9–10, line 199–226):
    "The summertime change characteristics of ground-level meteorological factors (MLH, RH, temperature, and WS), MDA8 $O_3$, $PM_{2.5}$ and its major components in the NCP were demonstrated in Figure 2 and Figure S1. The primary atmospheric pollutant in the NCP during the summertime was $O_3$, and the concentrations of MDA8

O$_3$ averaged over all sites in the NCP varied from 74.93 µg m$^{-3}$ to 219.28 µg m$^{-3}$, with the mean value of 151.72 µg m$^{-3}$ (Table R3). The O$_3$ pollution lasted nearly the whole observation period, characterized by frequent and long-lasting pollution episodes. The PM$_{2.5}$ concentration was much lower comparing with ozone, with the mean, maximum, and minimum of the regional daily averaged PM$_{2.5}$ concentration as 25.62 µg m$^{-3}$, 45.62 µg m$^{-3}$, and 11.32 µg m$^{-3}$, respectively. NO$_3^-$ was the prominent PM$_{2.5}$ component during summertime in the NCP, with the mean value of 7.76 µg m$^{-3}$. According to the National Ambient Air Quality Standard of China (GB3095-2012), the daily PM$_{2.5}$ averages in "2+26" cities can meet the Level II national ambient air quality standard (75 µg m$^{-3}$), while exceeding the level I standard of 35 µg m$^{-3}$. As showed in Figure 2, regional PM$_{2.5}$ pollution processes corresponded well with the increasing processes of MDA8 O$_3$. Here, we defined a O$_3$–PM$_{2.5}$ co-polluted episode as a set of continuous days (longer than 4 days) with MDA8 O$_3$ and daily mean PM$_{2.5}$ in more than 10 % NCP cities exceeding 160 µg m$^{-3}$ and 35 µg m$^{-3}$, respectively. According to this criterion, three typical O$_3$–PM$_{2.5}$ co-polluted episodes were selected: June 4–14 (Episode I), June 18–29 (Episode II), and July 2–11 (Episode III), 2021, and these three typical O$_3$–PM$_{2.5}$ co-polluted episodes have been pointed out in Figure 2 and Figure S1.

During these three typical episodes, the synchronous change characteristics of air pollutants and mixing layer meteorology were analyzed. In Episode I and II, when MLH higher than 2100 m, both MDA8 O$_3$ and PM$_{2.5}$ concentrations were low. Along with the reduction of MLH (from 1800–2100 m to 1200–1800 m), regional MDA8 O$_3$ and PM$_{2.5}$ concentration both gradually climbed up. When MLH fell in the range of 1200–1800 m, MDA8 O$_3$ concentration reached the maximum with about 80 % areas higher than 170 µg m$^{-3}$. We found that there is a lag time between the concentration peak of MDA8 O$_3$ and that of PM$_{2.5}$ along with the reduction of MLH. With the further decrease of MLH, MDA8 O$_3$ turned to decline, while PM$_{2.5}$ kept stable or continued to increase when regional MLH in the range of 600–1200 m. In Episode III, the MLH in most cities was lower than 1200 m, and the regional MDA8 O$_3$ and PM$_{2.5}$ pollution conditions were lighter than other episodes, with 80 % PM$_{2.5}$ values lower than 35 µg m$^{-3}$..."

In Section 3.2 and 3.3, we will comprehensively clarify the response of O$_3$, PM$_{2.5}$ and its secondary components to different mixing layer meteorology factors along the evolution of mixing layer.

**Comment 12:** Line 186, when stating "low boundary layer height," adding a value in parentheses would be helpful.

**Answer:** This information "high O$_3$ concentrations corresponded to a low boundary layer height" was cited by Zhao et al. (2019). However, we have failed to find the original reference (NASTRO, 2000) through full-text electronic periodicals database, such as Science Direct and Springer, and the exact value of "low boundary layer height" can not be obtained. Thus, we deleted this sentence in our revised manuscript.

**Comment 13:** In the sentence at 191, it presents a dichotomy between vertical transport and photochemical formation. Can't both factors happen simultaneously?

**Answer:** Yes, these two processes can happen simultaneously. This relationship observed between MDA8 $O_3$ and MLH is very complex. Previous works have shown that higher height of mixing layer can lead to the mixing of near surface air with the $O_3$ rich air aloft, resulting in the observed enhancements in surface $O_3$ concentration (Reddy et al., 2012). Concurrently, the evolution of mixing layer were strongly associated with the change of other meteorological conditions, such as air temperature, RH and precipitation, which can also affect $O_3$ concentration (Haman et al., 2014). The combined effects of these processes ultimately determine whether ground-level $O_3$ increases or not along the evolution of mixing layer. The above discussion has been added in our revised manuscript (Page 12, line 255–261).

**Comment 14:** Line 202, authors could add wet deposition for ozone as another elimination pathway, not only as a slowdown of photochemistry.

**Answer:** Thank you for valuable comments. The effect of wet deposition for ozone has been added in our revised manuscript: "due to higher RH or rain fall, gaseous precursors and $O_3$ can be washed out from the atmosphere trough wet deposition (Reddy et al., 2012)". (Page 12, line 267–268)

**Comment 15:** The paragraph starting at line 211 and ending at 236 needs to be rephrased. These are not results.

**Answer:** Thank you for valuable comments. The paragraph starting at line 211 and ending at 236 has been rephrased in our revised manuscript (Page 14, lines 287–300).

**Comment 16:** In the statement on line 240, it is true that SO4 and OC values reach their maximum at 900-1200m, but they are not significantly lower until 300-600m. How significant is this maximum? The same applies to NO3 and NH4 mentioned in line 242.
Line 249: "Changes in NH4 were a consequence of the changes in SO4 and NO3." It would be helpful to add the reason why; this sentence does not seem to be properly linked to the rest.

**Answer:** We have rephrased this paragraph in our revised manuscript (Page 15–16, line 308–318). The response of $PM_{2.5}$ concentrations to mixing layer structure was the net effect of the changes in $PM_{2.5}$ major chemical components. The purpose of this paragraph was to demonstrate the different change characteristics of these $PM_{2.5}$ components, and explore the intrinsic complexity mechanisms of $PM_{2.5}$ formation. These results showed that all the secondary components showed increasing trends when MLH changed from 0–300 m to 900–1200 m, with $SO_4^{2-}$ and OC showing the highest increment, followed by $NO_3^-$ and $NH_4^+$. When MLH changed from 300–600

m to 900−1200 m, the increment was not significant for $NO_3^-$ and $NH_4^+$. As $NH_3$ was generally abundantly supplied in the NCP, the formation of $NH_4^+$ was dominantly controlled by the reaction of ammonia with sulfate and nitrate aerosols, and the changes in $NH_4^+$ were a consequence of the changes in $SO_4^{2-}$ and $NO_3^-$ (Chow et al., 2022). When MLH<1200 m, the mass fraction of $NO_3^-$ was higher than $SO_4^{2-}$ in $PM_{2.5}$ (Figure S3), and the change characteristics of $NH_4^+$ along with the evolution of mixing layer were consistent with that of $NO_3^-$.

**Comment 17:** In line with the previous discussion, line 261, "competing changes in NO3 and SO4," does it mean that what decreases in one goes to the other?
I believe Figure 7 is mentioned before Figure 6.
In line 265, the phrase "the partitioning of semi-volatile nitrate is temperature-dependent" is not clear in relation to the previous information.

**Answer:** "Competing changes in $NO_3^-$ and $SO_4^{2-}$" means the opposite effect of temperature on the change characteristics of $NO_3^-$ and $SO_4^{2-}$. It's worthwhile noting that $SO_4^{2-}$ concentration climbed up with temperature increase. Higher temperature could promote faster oxidation of $SO_2$ to $SO_4^{2-}$, resulting in a significant increase in $SO_4^{2-}$ concentration. Unlike $SO_4^{2-}$, which predominantly exists in the particle phase, $NO_3^-$ could be either presented as nitric acid ($HNO_3$) in the gas phase or as ammonium nitrate ($NH_4NO_3$) in the particle phase (Chow et al., 2022). The temperature condition strongly influences the partitioning of nitrate between gas and particle phase. Higher temperature can prompt the partitioning of nitrate to $HNO_3$, thus nitrate tends to exit in the gas phase, resulting in a significant decrease in $NO_3^-$ and $NH_4^+$ concentrations. (Page 16, line 330−340)

**Comment 18:** The statement in line 270 "When MLH fell in the range..." should reference the figure. The same applies to what it is mentioned up to line 278.
 Lines 293 to 296: "Under low MLH..." Where is all this observed?
The paragraph starting at line 306 does not reference the figures, making it hard to understand.

**Answer:** Thank you for valuable comments. We have referenced the figures in our revised manuscript (Page 16−18, lines 344, 350, 361 and 375).

**Comment 19:** In line 291, the WHO guideline is suddenly mentioned. Why is it brought up all of a sudden?

**Answer:** The purpose was to demonstrate that event though the $PM_{2.5}$ concentration (31.34 $\mu g\ m^{-3}$) under moderate MLH condition did not exceeded the national air quality standard of $PM_{2.5}$ (daily mean of 35 $\mu g\ m^{-3}$), the $PM_{2.5}$ was still about two times higher than the WHO's $PM_{2.5}$ guideline value (daily mean of 15 $\mu g\ m^{-3}$, World Health Organization, 2021). This sentence does not seem to be properly linked to the

rest. Considering the readability of this paragraph, we have deleted this sentence in our revised manuscript.

**Comment 20:** Line 302: "The upward trend..." is not clear.

**Answer:** We have modified the description in our revised manuscript: "The increase of $PM_{2.5}$ and its major chemical components under medium MLH condition was not only associated with the weaker particle removal process by precipitation, but also related to the enhancement of secondary aerosol formation due to appropriate chemical reaction environment".(Page 17, line 369−371)

**Comment 21:** Concentrations of daily averages are also analyzed, but in this kind of episodes, the relevance of hourly concentrations can be crucial. This should also be made clear somewhere, although the analysis of the episodes is only carried out to highlight certain results.
In section 3.4, why is only one episode analyzed? If analyzing a single episode, it would be convenient to include a map with the synoptic evolution of those days.

**Answer:** In this work, we find a typical $PM_{2.5}$-$O_3$ coordinated event (Episode II: June 18–29, 2021) during the observation period to comprehensively present the relationship between the mixing layer meteorology and air pollutants. Figure R7 and R8 showed the temporal-spatial distribution of air pollutants and meteorological factors during June 18−29, 2021. On June 18−20, MLH gradually increased from 600−1200 m to 1500−3000 m in the southern and eastern part of the NCP, $PM_{2.5}$ and MDA8 $O_3$ concentrations concurrently increased and showed similar spatial distributions. The wind speed dropped significantly on 20 June, and the value was lower than 1 m s$^{-1}$ in most cities. On 21−23 June, MLH started to decrease from 1500−3000 m to 1200−1800 m, $PM_{2.5}$ and MDA8 $O_3$ concentrations further increased, and the areas of high $PM_{2.5}$ concentrations coincided well with those of MDA8 $O_3$ concentrations. During 24−25 June, MLH continued to decrease, with some values even lower than 300 m. The MLH for the areas with high MDA8 $O_3$ was in the range of 900−1500 m. Interestingly, the synchronized spatial change characteristics of $PM_{2.5}$ and MDA8 $O_3$ were consistent when MLH in the range of 900−1200 m, while inconsistent when MLH lower than 600 m. Significant rise of $PM_{2.5}$ concentration was observed in some cities with MLH lower than 300 m. It's noted that the dominant chemical composition of $PM_{2.5}$ in these areas was $NO_3^-$. On 28 June, the rise in MLH was observed in the central and the southern part in the NCP, and a surge of MDA8 $O_3$ and $PM_{2.5}$ concentrations both occurred, with 160−220 μg m$^{-3}$ and 40−50 μg m$^{-3}$ respectively. In general, most cities were dominated by weak winds from the east and southeast, which favored the formation of secondary pollutants from the gaseous precursors transported from the southeast part and promoted the accumulation of air pollutants.

To better understand this $PM_{2.5}$-$O_3$ co-polluted event, here we classified the observations during this typical event into four categories: $O_3$ polluted days ($O_3$PD;

MDA8 $O_3$ concentration $> 160\,\mu g\,m^{-3}$ and $PM_{2.5} < 35\,\mu g\,m^{-3}$), $PM_{2.5}$ polluted days ($PM_{2.5}$PD; MDA8 $O_3$ concentration $< 160\,\mu g\,m^{-3}$ and $PM_{2.5} > 35\,\mu g\,m^{-3}$), $O_3$–$PM_{2.5}$ co-polluted days ($O_3$–$PM_{2.5}$CPD; MDA8 $O_3$ concentration $> 160\,\mu g\,m^{-3}$ and $PM_{2.5} > 35\,\mu g\,m^{-3}$), and non-polluted days (NPD; MDA8 $O_3 < 80\,\mu g\,m^{-3}$ and $PM_{2.5} < 35\,\mu g\,m^{-3}$). Figure R9 showed the meteorological and chemical characteristic of $O_3$–$PM_{2.5}$ CPD, $O_3$ PD, $PM_{2.5}$ PD, and NPD. The results indicated that the values of MLH on $O_3$–$PM_{2.5}$CPD were between those on $O_3$PD and $PM_{2.5}$PD at around 900 m. On $O_3$–$PM_{2.5}$CPD, the oxidation ratio of sulfate (SOR, the molar ratio of sulfate to the sum of sulfate and $SO_2$) and oxidation ratio of nitrate (NOR, the molar ratio of nitrate to the sum of nitrate and $NO_2$) were the highest, with the values of 0.44 and 0.33, respectively, which indicated the strong secondary formation of $SO_4^{2-}$ and $NO_3^{-}$ promoted by high $O_3$ concentration. The $PM_{2.5}$PD occurred when MLH lower than 650 m, and the percentage of $NO_3^{-}$ was the highest on $PM_{2.5}$PD. The rise of $PM_{2.5}$ in some cities under low MLH conditions may be attributed to three mechanisms. The first one is the accumulation effect due to unfavorable diffusion condition when MLH decreased. Second, these cities got little rain, and the effect of wet deposition was weak. In addition, the corresponding low T and high RH can stimulate the formation of $NO_3^{-}$ from gaseous state ($HNO_3$). On $O_3$PD, the MLH was at around 1300 m, and the NOR turned to decrease, demonstrating a more significant role of partitioning process between gas and aerosol than the atmospheric oxidation process under this stage. On NPD, the MLH was the highest, with the value of about 2400 m, and the $PM_{2.5}$ chemical composition was obviously dominated by OM.

To explore the relevance of hourly $O_3$, $PM_{2.5}$, its components and MLH, we have taken PuY and HeZ as examples. Figure R10 plotted the day-to-day variations along with the diurnal variations of $O_3$, $PM_{2.5}$, its components and MLH in PuY and HeZ during Episode II (June 18−29, 2021). The results showed that there were large diurnal as well as day-to-day variability in the $O_3$ and $PM_{2.5}$ levels. The diurnal variations of MLH were clearly visible (Figure R11), with the rise in MLH during the daytime and the decrease in MLH at night. The concentration of $PM_{2.5}$ increased with the decrease of MLH at night, but the concentration of $O_3$ increased with the rise of MLH at daytime. Interestingly, we observed noontime soar of $SO_4^{2-}$ and OC concentrations in PuY, and the values of SOR kept stable or even increased at noon. Besides, it's noted that daily $O_3$ and $PM_{2.5}$ both gradually accumulated with the increase of MLH during June 18−21 and 26−28, which can be attributed to the $O_3$ and $PM_{2.5}$ superposition composite effects. The decrease in $PM_{2.5}$ at daytime with the rise of MLH can be partly offset by an increment in secondary pollutants formation derived from $O_3$ growth. Then with the decrease of MLH at night, the concentration of the original existing $PM_{2.5}$ increased due to unfavorable diffusion. In general, the conclusions in this work was only suitable to the day-to-day relationship between air pollutants and MLH. The hourly relationships were much more complicated and need more further analysis. According to your valuable comments, we will focus on the hourly relationship in the NCP in our follow-up studies. The above discussion has been added in our revised manuscript (Page 19−24, line 393−476).

**Comment 22:** When presenting the soundings, the interpretation of the data and the significance of the extinction coefficient should be explained. Why were soundings taken in those cities?

**Answer:** According to the valuable suggestion of Reviewer #2, the ozone radar has great shortcomings in the summer when there are clouds and high humidity, and the data may send wrong message and misled readers. Therefore, we deleted the soundings part in our revised manuscript.

**Comment 23:** In Figure 8, acronyms like OS and SO, do they mean the same thing?

**Answer:** No, the acronyms of OS and SO were different. The dominant $PM_{2.5}$ chemical component type was identified as follows: if the mass fraction of the maximum component was 1.2 times higher than that of the secondary one, the former was considered as the dominant factor, otherwise both dominated $PM_{2.5}$ formation. For the chemical component types of OS and SO, OM and $SO_4^{2-}$ both dominated $PM_{2.5}$ formation, however the maximum $PM_{2.5}$ component was different. For acronyms OS, OM was the maximum component, and $SO_4^{2-}$ was the secondary one; while as for SO, it is the other way around.

**Comment 24:** Linguistic issues in English to review: Expressions erroneously translated that result in: 'To the first order', 'as the MLH fixed', 'serious' (subjective), 'atmospheric dissipation (or dispersion?) potential', 'obvious (subjective)', 'the above analysis', 'MLH prefers'."

**Answer:** Thanks for your valuable comments. These mistakes have been corrected in our revised manuscript. (Line 61, 264, 332, 349, 363, 394, and 514)

**References**

Chow, W. S., Liao, K., Huang, X. H. H., Leung, K. F., Lau, A. K. H., and Yu, J. Z.: Measurement report: The 10-year trend of $PM_{2.5}$ major components and source tracers from 2008 to 2017 in an urban site of Hong Kong, China, Atmos. Chem. Phys., 22, 11557-11577, 10.5194/acp-22-11557-2022, 2022.

Chu, B., Ma, Q., Liu, J., Ma, J., Zhang, P., Chen, T., Feng, Q., Wang, C., Yang, N., Ma, H., Ma, J., Russell, A. G., and He, H.: Air Pollutant Correlations in China: Secondary Air Pollutant Responses to NOx and $SO_2$ Control, Environ. Sci. Technol. Let., 7, 695-700, 10.1021/acs.estlett.0c00403, 2020.

Dai, H., Liao, H., Li, K., Yue, X., Yang, Y., Zhu, J., Jin, J., Li, B., and Jiang, X.: Composited analyses of the chemical and physical characteristics of co-polluted days by ozone and $PM_{2.5}$ over 2013–2020 in the Beijing–Tianjin–Hebei region, Atmos. Chem. Phys., 23, 23-39, 10.5194/acp-23-23-2023, 2023.

Haman, C. L., Couzo, E., Flynn, J. H., Vizuete, W., Heffron, B., and Lefer, B. L.: Relationship between boundary layer heights and growth rates with ground-level ozone in Houston, Texas, J. Geophys. Res. Atmos., 119, 6230-6245, 10.1002/2013jd020473, 2014.

Haugen, D. A., Kaimal, J. C., Bradley, E. F.: An experimental study of Reynolds stress and heat flux in the atmospheric surface layer, Q. J. Roy. Meteor. Soc., 97, 168-180, 1971.

Li, M., Zhang, Z., Yao, Q., Wang, T., Xie, M., Li, S., Zhuang, B., and Han, Y.: Nonlinear responses of particulate nitrate to NOx emission controls in the megalopolises of China, Atmos. Chem. Phys., 21, 15135-15152, 10.5194/acp-21-15135-2021, 2021.

NASTRO (The North American Research Strategy for Tropospheric Ozone): An assessment of tropospheric ozone pollution: a North American perspective , 2000.

Niu, T., Wang, J., Yang, Y., Wang, Y., and Chen, C.: A study on parameterization of the Beijing winter heavy haze events associated with height of pollution mixing layer, Adv. Meteorol., 2017, 1-11, 10.1155/2017/8971236, 2017.

Reddy, K. K., Naja, M., Ojha, N., Mahesh, P., and Lal, S.: Influences of the boundary layer evolution on surface ozone variations at a tropical rural site in India, J. Earth Syst. Sci., 121, 911-922, 10.1007/s12040-012-0200-z, 2012.

Wang, J., Yang, Y.: Modern weather engineering. Meteorological Press, Beijing, 334–339, 2000.

Wang, J., Bian, L., Xiao, C.: Dynamics of ekman boundary layer over the antarctic plateau in summer, Chinese Sci. Bull., 59, 999–1005, 2014.

Wang, J., Yang, Y., Zhang, X., Liu, H., Che, H., Shen, X., and Wang, Y.: On the influence of atmospheric super-saturation layer on China's heavy haze-fog events, Atmos. Environ., 171, 261-271, 10.1016/j.atmosenv.2017.10.034, 2017.

Wang, J., Yang, Y., Jiang, X., Wang, D., Zhong, J., and Wang, Y.: Observational study of the $PM_{2.5}$ and $O_3$ superposition-composite pollution event during spring 2020 in Beijing associated with the water vapor conveyor belt in the northern hemisphere, Atmos. Environ., 272, 10.1016/j.atmosenv.2022.118966, 2022.

Wu, X., Xin, J., Zhang, W., Gao, W., Ma, Y., Ma, Y., Wen, T., Liu, Z., Hu, B., Wang,

Y., and Wang, L.: Variation characteristics of air combined pollution in Beijing City, Atmos. Res., 274, 10.1016/j.atmosres.2022.106197, 2022.

Zhao, W., Tang, G., Yu, H., Yang, Y., Wang, Y., Wang, L., An, J., Gao, W., Hu, B., Cheng, M., An, X., Li, X., and Wang, Y.: Evolution of boundary layer ozone in Shijiazhuang, a suburban site on the North China Plain, J Environ Sci (China), 83, 152-160, 10.1016/j.jes.2019.02.016, 2019.

Zhang, G., Bian, L., Wang, J., Yang, Y., Yao, W., Xu, X.: The boundary layer characteristics in the heavy fog formation process over Beijing and its adjacent areas, Sci. China Earth Sci., 48, 88–101, 2005.

**Table R1.** List of observation stations and locations.

| No. | Site | Abbreviation | Station | longitude (°E) | latitude (°N) |
|---|---|---|---|---|---|
| 1 | BeiJing | BeiJ | China National Environmental Monitoring Centre | 116.41 | 40.04 |
| 2 | Tianjin | TianJ | Zhongshan North Road Station | 117.21 | 39.17 |
| 3 | Shijiazhuang | SJZ | Northwest Shuiyuan Station | 114.49 | 38.13 |
| 4 | Langfang | LangF | Langfang Hebei University of Technology Station | 116.70 | 39.55 |
| 5 | Baoding | BaoD | Yangguang North Street Station | 115.48 | 38.93 |
| 6 | Tangshan | TangS | Xiaoshan Station | 118.19 | 39.62 |
| 7 | Handan | HanD | Guangming South Street Station | 114.50 | 36.57 |
| 8 | Hengshui | HengS | Hengshui Ecology and Environment Bureau Station | 115.68 | 37.73 |
| 9 | Xingtai | XingT | Quanbei Street Station | 114.53 | 37.09 |
| 10 | Cangzhou | CangZ | Cangzhou Technical College Station | 116.82 | 38.28 |
| 11 | Taiyuan | TaiY | Taiyuan Jinyuan Station | 112.48 | 37.71 |
| 12 | Yangquan | YangQ | Nanzhuang Road Station | 113.59 | 37.85 |
| 13 | Changzhi | ChangZ | Changzhi Ecology and Environment Bureau Station | 113.11 | 36.20 |
| 14 | Jincheng | JinC | Jincheng Ecology and Environment Bureau Station | 112.86 | 35.49 |
| 15 | Jinan | JiNan | Jinan Environmental Monitoring Station | 117.06 | 36.66 |
| 16 | Zibo | ZiB | Beijing Road station | 117.91 | 36.84 |
| 17 | Jining | JiNing | Jinyu Road Station | 116.63 | 35.43 |
| 18 | Dezhou | DeZ | Baima Lake Station | 115.83 | 36.95 |
| 19 | Liaocheng | LiaoC | Liaocheng monitoring center Station | 115.98 | 36.50 |
| 20 | Binzhou | BinZ | Binzhou Ecology and Environment Bureau Station | 118.01 | 37.38 |
| 21 | Heze | HeZ | Heze Quality Supervision | 115.53 | 35.21 |

| | | | Bureau Station | | |
|---|---|---|---|---|---|
| 22 | Zhengzhou | ZhengZ | Zhengzhou Forty-seven Middle School Station | 113.74 | 34.77 |
| 23 | Kaifeng | KaiF | Jinming West Street Station | 114.30 | 34.80 |
| 24 | Anyang | AnY | Anyang Ecology and Environment Bureau Station | 114.40 | 36.09 |
| 25 | Hebi | HeB | Hebi Ecology and Environment Bureau Station | 114.29 | 35.72 |
| 26 | Xinxiang | XinX | Xinxiang Ecology and Environment Bureau Station | 113.92 | 35.30 |
| 27 | Jiaozuo | JiaoZ | Fengshou Middle Road Station | 113.28 | 35.21 |
| 28 | Puyang | PuY | Jinti Road Station | 115.04 | 35.76 |

**Table R2.** List of observation species and metrics.

| Species | Unit | Temporal resolution | Metrics |
|---|---|---|---|
| Gaseous pollutants | | | |
| $O_3$ | $\mu g\ m^{-3}$ | 1 h | Maximum daily 8 h average |
| $SO_2$ | $\mu g\ m^{-3}$ | 1 h | Daily average |
| $NO_2$ | $\mu g\ m^{-3}$ | 1 h | Daily average |
| $PM_{2.5}$ and its major components | | | |
| $PM_{2.5}$ | $\mu g\ m^{-3}$ | 1 h | Daily average |
| $SO_4^{2-}$/ $NO_3^-$/ $NH_4^+$ | $\mu g\ m^{-3}$ | 1 h | Daily average |
| OC | $\mu g\ m^{-3}$ | 1 h | Daily average |
| Meteorological variables | | | |
| Temperature (T) | °C | 1 h | Daily average |
| Relative Humidity (RH) | % | 1 h | Daily average |
| Wind speed (WS) | $m\ s^{-1}$ | 1 h | Daily average |
| Wind direction (WD) | ° | 1 h | Daily average |
| 24-h precipitation | mm | 24 h | 24-h accumulated |

**Table R3.** General information on $O_3$–$PM_{2.5}$ co-polluted episodes from June 1 to July 31, 2021.

| | Episode I | | | Episode II | | | Episode III | | | Summer | | |
|---|---|---|---|---|---|---|---|---|---|---|---|---|
| | Ave. | Min | Max | Ave. | Min | Max | Ave. | Min | Max | Ave. | Min | Max |
| **Gaseous pollutants ($\mu g\ m^{-3}$)** | | | | | | | | | | | | |
| MDA8 $O_3$ | 170.80 | 85.62 | 219.28 | 180.65 | 142.10 | 204.15 | 168.70 | 111.79 | 199.39 | 151.72 | 74.94 | 219.28 |
| $SO_2$ | 10.01 | 6.48 | 14.44 | 9.09 | 6.11 | 12.48 | 6.75 | 5.72 | 8.00 | 7.59 | 4.79 | 14.44 |
| $NO_2$ | 24.61 | 16.26 | 31.81 | 22.89 | 14.11 | 32.15 | 17.66 | 13.12 | 21.00 | 19.31 | 10.90 | 32.15 |
| **$PM_{2.5}$ and its major components ($\mu g\ m^{-3}$)** | | | | | | | | | | | | |
| $PM_{2.5}$ | 30.55 | 15.74 | 42.67 | 28.33 | 17.22 | 42.52 | 25.05 | 20.84 | 31.75 | 25.62 | 11.32 | 45.62 |
| $NO_3^-$ | 8.74 | 2.16 | 16.44 | 8.29 | 2.85 | 18.00 | 7.67 | 5.87 | 13.44 | 7.76 | 2.16 | 18.24 |
| $SO_4^{2-}$ | 7.22 | 2.81 | 10.25 | 7.32 | 4.02 | 12.15 | 7.12 | 5.48 | 8.92 | 7.04 | 2.81 | 12.15 |
| $NH_4^+$ | 5.51 | 1.42 | 9.34 | 5.52 | 2.27 | 9.29 | 5.38 | 4.46 | 8.21 | 5.30 | 1.42 | 9.88 |
| OC | 5.11 | 2.74 | 6.60 | 4.71 | 3.25 | 6.75 | 4.11 | 2.90 | 5.30 | 4.32 | 2.69 | 6.75 |
| **Meteorological variables** | | | | | | | | | | | | |
| MLH (m) | 1342.73 | 305.93 | 2423.42 | 1190.36 | 626.51 | 2127.31 | 740.86 | 460.91 | 950.10 | 855.99 | 305.93 | 2423.42 |
| T (°C) | 26.24 | 23.86 | 28.91 | 27.41 | 25.53 | 28.76 | 27.58 | 24.85 | 30.14 | 26.69 | 22.48 | 30.14 |
| RH (%) | 57.01 | 32.78 | 90.54 | 56.90 | 37.04 | 70.60 | 71.45 | 64.64 | 80.38 | 68.70 | 32.78 | 90.54 |

[Figure]

**Figure R1.** The graphical abstract.

[Figure]

**Figure R2** The overall WS and WD condition during the observation campaign. S: south; N: north; E: east; W: west.

[Figure]

**Figure R3** The change characteristics of WS and WD under different MLH levels. S: south; N: north; E: east; W: west.

[Figure]

**Figure R4.** The distribution characteristics of the MDA8 O$_3$ concentrations with the evolution of MLH under different (a) temperature, (b) RH, (c) precipitation, (d) WS, and (e) WD conditions.

[Figure]

**Figure R5.** The distribution characteristics of the PM$_{2.5}$ concentrations with the evolution of MLH under different (a) temperature, (b) RH, (c) precipitation, (d) WS, and (e) WD conditions.

[Figure]

**Figure R6.** The distribution characteristics of $NO_3^-$, $SO_4^{2-}$, $NH_4^+$, and OC concentrations with the evolution of MLH under different (a) temperature, (b) RH, (c) precipitation, (d) WS, and (e) WD conditions.

[Figure]

**Figure R7.** The spatial distribution of (a) MLH, (b) MDA8 O$_3$, (c) PM$_{2.5}$, (d) the dominant PM$_{2.5}$ chemical component (N: NO$_3^-$ dominant, NS: NO$_3^-$ and SO$_4^{2-}$ dominant, NO: NO$_3^-$ and OM dominant, S: SO$_4^{2-}$ dominant, SN: SO$_4^{2-}$ and NO$_3^-$ dominant, SO: SO$_4^{2-}$ and OM dominant, O: OM dominant, ON: OM and NO$_3^-$ dominant, OS: OM and SO$_4^{2-}$ dominant), (e) T, and (f) RH, (g) the overall change characteristics of WS and WD in the NCP from June 18 to 23, 2021.

[Figure]

**Figure R8.** The spatial distribution of (a) MLH, (b) MDA8 $O_3$, (c) $PM_{2.5}$, (d) the dominant $PM_{2.5}$ chemical component (N: $NO_3^-$ dominant, NS: $NO_3^-$ and $SO_4^{2-}$ dominant, NO: $NO_3^-$ and OM dominant, S: $SO_4^{2-}$ dominant, SN: $SO_4^{2-}$ and $NO_3^-$ dominant, SO: $SO_4^{2-}$ and OM dominant, O: OM dominant, ON: OM and $NO_3^-$ dominant, OS: OM and $SO_4^{2-}$ dominant), (e) T, and (f) RH, (g) the overall change characteristics of WS and WD in the NCP from June 24 to 29, 2021.

[Figure]

**Figure R9.** The distribution characteristics of (a) NOR and SOR, and (b) the mass fractions of major $PM_{2.5}$ components, MLH, RH, and T under $O_3$–$PM_{2.5}$ CPD, $O_3$ PD, $PM_{2.5}$ PD, and NPD conditions from June 24 to 29, 2021.

[Figure]

**Figure R10.** The hourly evolution of $O_3$, $PM_{2.5}$, its components and MLH in HeZ and PuY during June 18−29, 2021.

[Figure]

**Figure R11.** The diurnal variation of MLH, SOR, NOR, O₃, PM₂.₅, and its components in HeZ and PuY during June 18−29, 2021.

---

## Author Comment (AC3)

Dear Professor Xavier Querol,

Thank you very much for handling our manuscript submitted to *Atmospheric Chemistry and Physics* (**MS No.:** egusphere-2023-479; **Title:** Summertime response of ozone and fine particulate matter to mixing layer meteorology over the North China Plain).

We deeply thank the reviewers for giving constructive comments and suggestions that are very helpful to improve our manuscript. The comments raised by the reviewers mostly focus on five aspects: 1) observation data profile description; 2) calculation method of occurrence frequency and MLH; 3) a case study of the typical $PM_{2.5}$-$O_3$ co-polluted episode; 4) influence of WS and WD; 5) limitation of this work. We have answered these comments point by point and listed in the following pages.

On behalf of all the co-authors, I would like to thank you and referees for all the invaluable comments. Please feel free to contact me if you need any further information.

Yours Sincerely,

Jian Gao

Professor

Chinese Research Academy of Environmental Sciences, Beijing 100012, China

E-mail address: gaojian@craes.org.cn

Phone: 18911819868

**1. Observation data profile description**

**Reviewer #2:**Lack of data profile description. For example, how many observatories are there, and what observation elements do each station have? The author should at least add a table that fully illustrates the data.

**Reviewer #1:**In the map of Figure 1, it is advisable to specify the exact locations of the measurements. Are they conducted within cities? What environments have been selected, and what criteria were used for their selection? Is there any measurement that could be heavily influenced by local emissions?

Line 106, the meteorological variables are measured at the same measurement site, right?

What is the temporal resolution of the meteorological measurements? Have they been averaged to obtain daily averages? It is not clear.

**Answer:** Our field observations were conducted based on the existing ground-level observation stations (national control station, $PM_{2.5}$ component network, etc.) in the North China Plain (NCP), which covered two megacities (BeiJ and TianJ) and 26 surrounding cities. The spatial distribution of these 28 valid sites was shown in Figure 1. Hourly concentrations of ground-level $SO_2$, $NO_2$, $O_3$, $PM_{2.5}$ and its chemical compositions ($SO_4^{2-}$, $NO_3^-$, $NH_4^+$, and OC), and meteorological variables, including air temperature, relative humidity (RH), wind speed (WS) and direction (WD), and 24-h accumulated precipitation, at the sites were obtained from the platform of National Atmospheric Particulate Chemical-Speciation-Network. This network is established to improve the understanding of the heavy pollution formation mechanism in the North China Plain (NCP) and support the decision-making of local governments and state administration. Hourly $SO_2$, $NO_2$, $O_3$, $PM_{2.5}$ and its chemical compositions were recorded at the $PM_{2.5}$ component network, which was selected followed the Technical Regulation for Selection of Ambient Air Quality Monitoring Station published by the Ministry of Ecology and Environment of the People's Republic of China (HJ664−2013). The monitoring sites of $PM_{2.5}$ component network were mostly set up within the cities, and can reflect the average pollution level of each city. Details for the near-ground observation stations of $PM_{2.5}$ component network were shown in Table R1. The meteorological variables were recorded in the national meteorological observation stations, and the information of each station can be obtained from the public website of China Meteorological Administration (http://data.cma.cn/data/cdcindex/cid/0b9164954813c573.html). It should be noted that the measurement sites of meteorological variables and air pollutants were not always consistent. To better analyze the meteorological conditions for $O_3$ and $PM_{2.5}$, only the station closed to the air quality monitoring station and representative of the city meteorological condition was selected in our work. The temporal resolution of air temperature, RH, WS and WD was 1-h. To avoid the influence of diurnal boundary layer cycles, in this article we focused on the relationships between daily mean air pollutants and meteorological factors. The daily mean meteorological factors, $PM_{2.5}$ and its major secondary components were calculated from the hourly data; daily $O_3$ concentration was characterized by the maximum daily 8 h average ozone (MDA8

O$_3$). Details for the near-ground observation species and the metrics were shown in Table R2. The above discussion has been added in our revised manuscript **(Page 6−7, line 111−143)**.

**Table R1.** List of observation stations and locations.

| No. | Site | Abbreviation | Station | longitude (°E) | latitude (°N) |
|---|---|---|---|---|---|
| 1 | BeiJing | BeiJ | China National Environmental Monitoring Centre | 116.41 | 40.04 |
| 2 | Tianjin | TianJ | Zhongshan North Road Station | 117.21 | 39.17 |
| 3 | Shijiazhuang | SJZ | Northwest Shuiyuan Station | 114.49 | 38.13 |
| 4 | Langfang | LangF | Langfang Hebei University of Technology Station | 116.70 | 39.55 |
| 5 | Baoding | BaoD | Yangguang North Street Station | 115.48 | 38.93 |
| 6 | Tangshan | TangS | Xiaoshan Station | 118.19 | 39.62 |
| 7 | Handan | HanD | Guangming South Street Station | 114.50 | 36.57 |
| 8 | Hengshui | HengS | Hengshui Ecology and Environment Bureau Station | 115.68 | 37.73 |
| 9 | Xingtai | XingT | Quanbei Street Station | 114.53 | 37.09 |
| 10 | Cangzhou | CangZ | Cangzhou Technical College Station | 116.82 | 38.28 |
| 11 | Taiyuan | TaiY | Taiyuan Jinyuan Station | 112.48 | 37.71 |
| 12 | Yangquan | YangQ | Nanzhuang Road Station | 113.59 | 37.85 |
| 13 | Changzhi | ChangZ | Changzhi Ecology and Environment Bureau Station | 113.11 | 36.20 |
| 14 | Jincheng | JinC | Jincheng Ecology and Environment Bureau Station | 112.86 | 35.49 |
| 15 | Jinan | JiNan | Jinan Environmental Monitoring Station | 117.06 | 36.66 |
| 16 | Zibo | ZiB | Beijing Road station | 117.91 | 36.84 |
| 17 | Jining | JiNing | Jinyu Road Station | 116.63 | 35.43 |
| 18 | Dezhou | DeZ | Baima Lake Station | 115.83 | 36.95 |
| 19 | Liaocheng | LiaoC | Liaocheng monitoring center Station | 115.98 | 36.50 |

| | | | | | |
|---|---|---|---|---|---|
| 20 | Binzhou | BinZ | Binzhou Ecology and Environment Bureau Station | 118.01 | 37.38 |
| 21 | Heze | HeZ | Heze Quality Supervision Bureau Station | 115.53 | 35.21 |
| 22 | Zhengzhou | ZhengZ | Zhengzhou Forty-seven Middle School Station | 113.74 | 34.77 |
| 23 | Kaifeng | KaiF | Jinming West Street Station | 114.30 | 34.80 |
| 24 | Anyang | AnY | Anyang Ecology and Environment Bureau Station | 114.40 | 36.09 |
| 25 | Hebi | HeB | Hebi Ecology and Environment Bureau Station | 114.29 | 35.72 |
| 26 | Xinxiang | XinX | Xinxiang Ecology and Environment Bureau Station | 113.92 | 35.30 |
| 27 | Jiaozuo | JiaoZ | Fengshou Middle Road Station | 113.28 | 35.21 |
| 28 | Puyang | PuY | Jinti Road Station | 115.04 | 35.76 |

**Table R2.** List of observation species and metrics.

| Species | Unit | Temporal resolution | Metrics |
|---|---|---|---|
| Gaseous pollutants | | | |
| $O_3$ | $\mu g\ m^{-3}$ | 1 h | Maximum daily 8 h average |
| $SO_2$ | $\mu g\ m^{-3}$ | 1 h | Daily average |
| $NO_2$ | $\mu g\ m^{-3}$ | 1 h | Daily average |
| $PM_{2.5}$ and its major components | | | |
| $PM_{2.5}$ | $\mu g\ m^{-3}$ | 1 h | Daily average |
| $SO_4^{2-}$/ $NO_3^-$/ $NH_4^+$ | $\mu g\ m^{-3}$ | 1 h | Daily average |
| OC | $\mu g\ m^{-3}$ | 1 h | Daily average |
| Meteorological variables | | | |
| Temperature (T) | ° C | 1 h | Daily average |
| Relative Humidity (RH) | % | 1 h | Daily average |
| Wind speed (WS) | $m\ s^{-1}$ | 1 h | Daily average |
| Wind direction (WD) | ° | 1 h | Daily average |
| 24-h precipitation | mm | 24 h | 24-h accumulated |

**2. Calculation method of the occurrence frequency**

**Reviewer #2:** Figure 2 is confusing. My understanding is that the proportion of different values occurring at a certain time should sum to 100%. But the sum expressed in the figure must be more than 100%.

**Reviewer #1:** The % ratio of occurrence shown in Figure 2 should be mentioned in the methodology. How has it been calculated?

**Answer:** We have added the calculation method of the occurrence frequency (%) mentioned in Figure 2 in the methodology **(Page 7, line 144−150)**. To better demonstrate the overall change characteristics of regional air pollution and meteorological conditions during the observation period, the occurrence frequency (%), which means the proportion of the number of cities at each air pollutant or methodology level, was calculated based on the following equation:

$$\text{Occurrence frequency}_X^{level} = \frac{N_X^{level}}{\text{Total } N_X} \times 100\% \tag{1}$$

where X means the air pollutant or methodology factors, $N_X^{level}$ represents the number of cities at each X level, $\text{Total } N_X$ represents the total number of cities. For example, as for the MLH condition, the MLHs were classified into 8 levels, and this ratio indicates the proportion of the number of cities at each MLH level to the total number of cites. As can be seen in Figure 2, on June 5, 2021, the proportion of the number of cities at MLH>2100 m was around 85 %, and significantly higher than other MLH conditions; on June 10, 2021, the MLH in all cities were lower than 1200 m, with the ratio at MLH<1200 as 100%.

**3. Calculation method of MLH**

**Reviewer #2:** In view of the importance of MLH, the authors have not confirmed the calculation results. The authors use a very simple method to calculate the height of the mixed layer. It is suggested that the author make use of the meteorological profile or ERA5 reanalysis data to verify the reliability of the results.

Generally, when the boundary layer rises, the wind speed will increase, especially when the boundary layer exceeds 1500 m, but this phenomenon is not shown in this study. In addition, in general, when precipitation occurs, strong convection occurs, and the height of the mixing layer will suddenly rise, which is different from the author's study. Therefore, it is highly recommended that the authors validate the results of mixing layer height.

**Reviewer #1:** Point 2.2. The methodology for calculating the MLH seems somewhat simplistic. While estimating MLH can be complex, it would be advisable to compare the results with sounding data (already shown) or reanalysis data to determine their coherence with the calculated values.

**Answer:** Even though the method for calculating MLH in this work seems simple, this methodology reflects the basic physical nature of the pollution mixing layer height. In recent years, many works have progressed in the atmospheric boundary layer characteristics, and analyzed the impacts of these parameter on air pollution. Planetary boundary layer (PBL), as one of the critical parameters to air quality modeling, has been well explored. However, PBL usually refers to the large-scale Ekman dynamic boundary layer (Haugen et al., 1971; Wang et al., 2014; Zhang et al., 2005). The way with which boundary layer describes the influences of air pollution is

easily duplicated and confused (Niu et al., 2017). It is unreasonable to some extent, if the characteristic of the air pollution related to near-surface boundary layer is evaluated by using the concept of PBL. For air pollution measurement, one of selected functionalities of parameterization scheme for pollution mixing layer is to judge whether an air mass over a specific locality satisfies the "static and stable" attribute or not. Therefore, in this work, to express the basic physics for diagnosing meteorological conditions, we used the concept of pollution mixing layer height (MLH) proposed by Wang et al. (2017), which was based on the classical synoptic theory according to the level of convective condensation layer, and the details of this method can be seen in previous work (Wang and Yang, 2000; Wang et al., 2017).

To be specific, we defined the height close to the cloud base as the height of super-saturation layer (H_SSL), and the isoentropic atmospheric process meets the level of convective condensation layer (LCL) in the super-saturation state, i.e., it is very close to the H_SSL. Iterative algorithm is used to work out the H_SSL (Wang and Yang, 2000):

$$H\_SSL \approx LCL = 6.11 \times 10^2 \times \left( \frac{0.622 + 0.622 \frac{e_s}{p - e_s}}{0.622 \frac{e_s}{p - e_s}} \right) \tag{2}$$

$$e_s = 6.22 \times \exp \frac{17.13(T - 273.16)}{T - 38} \tag{3}$$

where $e_s$ represents saturated water vapor pressure, T is temperature (K). Eq. (2) can be used to calculate the H_SSL which is favorable for pollutant mixing and represented by (P). Below this height, the atmosphere gets supersaturated, causing the pollution mixing and wetting process in the low altitude to continue, so this height is also called the height of pollution mixing layer (MLH). Thus, MLH can be derived in the following expression:

$$MLH \approx H\_SSL \approx LCL = 6.11 \times 10^2 \times \left( \frac{0.622 + 0.622 \frac{e_s}{p - e_s}}{0.622 \frac{e_s}{p - e_s}} \right) \tag{4}$$

According to the relationship between air pressure and height, the units of MLH can be converted to the height expression in meters:

$$\int_{p_0}^{p_z} dp = - \int_0^z \rho_0 g \, dz, \tag{5}$$

where z is the height, $\rho_0$ is the density of gas, $p_z$ and $p_0$ represent the air pressure in the height of z and 0, respectively.

Several works have verified the reliability of the results based on this method. With this method, Wang et al. (2017) well characterized the features of mixing layer height in highly-sensitive areas of pollution in China ( 1−31 December 2015 for Beijing and the same period of 1−31 December 2015 for Guangzhou), and demonstrated the schematic diagram of 3-D model for low-level super-saturation layer and pollution mixing layer in the pollution hotspots in China, such as North China Plain (NCP), Yangtze River Delta (YRD), Pearl River Delta (PRD) and Si-Chuan Basin (SCB). Wang et al. (2022) also used this method to explore the $PM_{2.5}$ and $O_3$ superposition-composite pollution event during spring 2020 in Beijing, China,

and the hourly evolution of MLH, $O_3$, and $PM_{2.5}$ during the observation period were analyzed. The results can well depict the MLH diurnal cycle, which rises at daytime and decreases at night. In addition, Niu et al. (2017) has applied this method in Beijing, and the results showed that the pollution mixing layer can well present the change characteristics of haze pollution process. In this work, we further clarified the concept of MLH, and applied this method to investigate the impacts of MLH upon the change characteristics of ozone and fine particulate matter. The above discussion has been added in our revised manuscript **(Page 7−8, line 152−190)**.

As we known, there are differences between MLH and PBLH (Height of planetary Boundary layer). These phenomena ("when the boundary layer rises, the wind speed will increase, especially when the boundary layer exceeds 1500 m", "when precipitation occurs, strong convection occurs, and the height of the mixing layer will suddenly rise") were generally summarized based on PBLH in individual cases. Besides, these phenomena can not fit each case, and there are still exceptions. We have added the change characteristics of wind speed (WS) along with the increase of MLH (Figure R1). Actually, we can see apparent increase of WS when MLH in the range of 0−300 m which was probably due to precipitation events. The increase of WS when MLH exceeds 1500 have also been observed, but the increment was not so obvious. Previous works by Liu and Liang (2010) and Li et al. (2020) have found that the severe convective weather generally decreases PBLH, and the precipitation was highly negatively correlated with PBLH, which was consistent with the results found in our work. The rainfall events may produce clouds, then reduce surface solar and thermal heating, thus suppressing the PBLH.

[Figure]

**Figure R1.** The change characteristics of WS under different MLH conditions.

**4. A case study of the typical $PM_{2.5}$-$O_3$ co-polluted episode**

**Reviewer #2:** The authors need to find a case to fully present the relationship between the mixing layer and pollutants, and use the hourly concentration to illustrate the response of PM, its components and ozone to the mixing layer.

**Reviewer #1:** Concentrations of daily averages are also analyzed, but in this kind of

episodes, the relevance of hourly concentrations can be crucial. This should also be made clear somewhere, although the analysis of the episodes is only carried out to highlight certain results.

**Answer:** We find a typical $PM_{2.5}$-$O_3$ coordinated event (Episode II: June 18–29, 2021) during the observation period to comprehensively present the relationship between the mixing layer meteorology and air pollutants. Figure R2 and R3 showed the temporal-spatial distribution of air pollutants and meteorological factors during June 18−29, 2021. On June 18−20, MLH gradually increased from 600−1200 m to 1500−3000 m in the southern and eastern part of the NCP, $PM_{2.5}$ and MDA8 $O_3$ concentrations concurrently increased and showed similar spatial distributions. The wind speed dropped significantly on 20 June, and the value was lower than 1 m s$^{-1}$ in most cities. On 21−23 June, MLH started to decrease from 1500−3000 m to 1200−1800 m, $PM_{2.5}$ and MDA8 $O_3$ concentrations further increased, and the areas of high $PM_{2.5}$ concentrations coincided well with those of MDA8 $O_3$ concentrations. During 24−25 June, MLH continued to decrease, with some values even lower than 300 m. The MLH for the areas with high MDA8 $O_3$ was in the range of 900−1500 m. Interestingly, the synchronized spatial change characteristics of $PM_{2.5}$ and MDA8 $O_3$ were consistent when MLH in the range of 900−1200 m, while inconsistent when MLH lower than 600 m. Significant rise of $PM_{2.5}$ concentration was observed in some cities with MLH lower than 300 m. It's noted that the dominant chemical composition of $PM_{2.5}$ in these areas was $NO_3^-$. On 28 June, the rise in MLH was observed in the central and the southern part in the NCP, and a surge of MDA8 $O_3$ and $PM_{2.5}$ concentrations both occurred, with 160−220 µg m$^{-3}$ and 40−50 µg m$^{-3}$ respectively. In general, most cities were dominated by weak winds from the east and southeast, which favored the formation of secondary pollutants from the gaseous precursors transported from the southeast part and promoted the accumulation of air pollutants.

[revised manuscript text omitted]

**Figure R5.** The hourly evolution of O₃, PM₂.₅, its components and MLH in HeZ and PuY during June 18−29, 2021.

[Figure]

**Figure R6.** The diurnal variation of MLH, SOR, NOR, O₃, PM₂.₅, and its components in HeZ and PuY during June 18−29, 2021.

**5. The influence of WS (Wind Speed) and Wind Direction (WD)**

**Reviewer #1:** The WS (Wind Speed) is barely mentioned, and I miss the Wind Direction. Although the authors discard advection, how can authors be certain that it is not always due to the same emission source or the same synoptic pattern?

**Answer:** To reveal the impact of wind on the variation trends of air pollution, we presented the results of wind speed (WS) and wind direction (WD) in our revised manuscript (Figure R7– R11). As shown in Figure R7, during the whole campaign, the NCP was dominated by winds from northeast and south (45–225°). Because more than 75 % WD were in the rang of 45–225°, the WD was classified into 4 categories: 45–90, 90–135, 135–180, and 180–225°. To further investigate the impact of wind on MDA8 $O_3$ and $PM_{2.5}$ concentrations, we also demonstrated the statistics on the concentration distributions of MDA8 $O_3$, $PM_{2.5}$ and its dominant components with the increase of MLH under different WS and WD conditions in Figure R9–R11. In general, WS could affect the diffusion of air pollutants. During the observation period, WS was mostly less than 3 m s$^{-1}$, and the concentrations of air pollutants were comparatively higher at low wind speeds. As shown in Figure R8, at low MLH conditions (MLH<300 m), a northeasterly wind prevailed near the ground, and the WS was generally higher than other conditions. The concentration of MDA8 $O_3$ was low during this period. With the increase of MLH, the WD gradually changed from northeast (MLH=300–600 m) to southeast (MLH=600–900 m) and south (MLH=900–1200 m). The North China Plain (NCP) is surrounded to the west by the Taihang Mountains, to the north by the Yan Mountains, and to the east by the Bohai Sea (Figure 1). The southerly wind can transport the gaseous pollutant or $O_3$ from the southern part of the plain area to the northern part, and the Taihang mountains may block pollutant transport, leading to the accumulation of pollutants along the foot of the Taihang Mountains. It's noted that the concentration of MDA8 $O_3$ was higher when the plain dominated by southerlies (180–225°) (Figure R9). When MLH higher than 2100 m, NCP was governed by northwest winds, which preferred the outward transport and diffusion of pollution, leading to the decrease of MDA8 $O_3$. Comparing with $O_3$, the impact of WD along with the increase of MLH seems different for $PM_{2.5}$ and its dominant components. When MLH in the range of 600–1200 m, the NCP was dominated by southeast or south winds. However, when southeast or south wind prevailed, the corresponding $PM_{2.5}$ and its dominant components concentrations were comparable or even lower than other WD situations (Figure R10–R11). This indicated that regional transport was not the dominant factor leading to the elevation of $PM_{2.5}$ and its aerosol species along with the evolution of mixing layer (MLH < 1200 m). The above discussion has been added in our revised manuscript **(Page 12−13, line 271−283; Page 18, line 377−383;)**.

[Figure]

**Figure R7** The overall WS and WD condition during the observation campaign. S: south; N: north; E: east; W: west.

[Figure]

**Figure R8** The change characteristics of WS and WD under different MLH levels. S: south; N: north; E: east; W: west.

[Figure]

**Figure R9.** The distribution characteristics of the MDA8 O₃ concentrations with the evolution of MLH under different (a) temperature, (b) RH, (c) precipitation, (d) WS, and (e) WD conditions.

[Figure]

**Figure R10.** The distribution characteristics of the PM$_{2.5}$ concentrations with the evolution of MLH under different (a) temperature, (b) RH, (c) precipitation, (d) WS, and (e) WD conditions.

[Figure]

**Figure R11.** The distribution characteristics of $NO_3^-$, $SO_4^{2-}$, $NH_4^+$, and OC concentrations with the evolution of MLH under different (a) temperature, (b) RH, (c) precipitation, (d) WS, and (e) WD conditions.

**6. Limitation of this study**

**Reviewer #1:** There is a noticeable absence of indication regarding potential limitations and uncertainties in this study.

The analyzed results are limited to only two summer months, which may not be easily extrapolated to other summers. It is essential to address why the three mentioned episodes are relevant and whether the selected two months are representative of typical patterns observed throughout the years. Were there any meteorological anomalies during this period? These aspects should clear throughout the entire manuscript to provide a more comprehensive understanding of the study's scope and implications.

**Answer:** According to the hourly concentrations of $PM_{2.5}$ and MDA8 $O_3$ in China over the years of 2013–2020, the observed numbers of $O_3$ polluted days (MDA8 $O_3$ concentration $> 160 \, \mu g \, m^{-3}$ and $PM_{2.5} < 75 \, \mu g \, m^{-3}$), $PM_{2.5}$ polluted days (MDA8

$O_3$ concentration $< 160\,\mu g\,m^{-3}$ and $PM_{2.5} > 75\,\mu g\,m^{-3}$), and $O_3$–$PM_{2.5}$ co-polluted days (MDA8 $O_3$ concentration $> 160\,\mu g\,m^{-3}$ and $PM_{2.5} > 75\,\mu g\,m^{-3}$) were highest in the North China Plain (NCP). Besides, $O_3$–$PM_{2.5}$ co-polluted days and $O_3$ polluted days were generally occurred in June and July (Dai et al., 2023). Therefore, we considered the months of June and July can well represent the typical characteristics of $O_3$–$PM_{2.5}$ coordinated pollution during warm seasons in the North China Plain (NCP). According to the National Ambient Air Quality Standard of China (GB3095-2012), the daily $PM_{2.5}$ averages in "2+26" cities can meet the Level II national ambient air quality standard ($75\,\mu g\,m^{-3}$), while exceeding the level I standard of $35\,\mu g\,m^{-3}$. Here, we defined a $O_3$–$PM_{2.5}$ co-polluted episode as a set of continuous days (longer than 4 days) with MDA8 $O_3$ and daily mean $PM_{2.5}$ in more than 10 % NCP cities exceeding $160$ $\mu g$ $m^{-3}$ and $35$ $\mu g$ $m^{-3}$, respectively. According to this criterion, three typical $O_3$–$PM_{2.5}$ co-polluted episodes were selected: June 4–14 (Episode I), June 18–29 (Episode II), and July 2–11 (Episode III), 2021. These three episodes have been marked in Figure 2 in our revised manuscript. Comparing with the same observation period in 2020, the temperature was slightly higher and more precipitation evens happened during the summertime in 2021 (National Bulletin of Atmospheric Environment, 2021; http://www.nmc.cn/publish/environment/National-Bulletin-atmospheric-environment. htm). This work can provide an overall diagnosis of the response of ozone and fine particulate matter to mixing layer meteorology during summertime, especially in warm and humid seasons, in the NCP.

This work can gain new insights into the underlying causes of the summertime $O_3$–$PM_{2.5}$ coordinated pollution through exploring the response of ozone and fine particulate matter to mixing layer meteorology over the North China Plain from June 1 to July 31, 2021. However, there remained some limitations and uncertainties. First, the present study was only confined to summertime conditions (including two summer months) in the NCP, and the conclusions was likely to be different in other seasons and regions. Thus, more extended observations in time and space should be needed in the future. Second, to avoid the influence of diurnal boundary layer cycles, in this article we focused on the relationships between daily mean air pollutants and meteorological factors. We have also taken PuY and HeZ as examples to explore the relevance of hourly $O_3$, $PM_{2.5}$, its components and MLH. The results showed large diurnal as well as day-to-day variability in the $O_3$ and $PM_{2.5}$ levels. The decrease in $PM_{2.5}$ at daytime with the rise of MLH can be partly offset by an increment in secondary pollutants formation derived from $O_3$ growth. In general, the conclusions in this work was only suitable to the day-to-day relationship between air pollutants and MLH. The hourly relationships were much more complicated and need more further analysis. Lastly, a weakness of this study is that we did not quantify the sensitivity of $O_3$ and $PM_{2.5}$ to different meteorological factors and chemical processes, thus a more detailed consideration with the aid of modeling would be needed in the future. Such an exploration requires efforts going beyond the current project, and it is therefore not pursued. The above discussion has been added in our revised manuscript. **(Page 21, line 449−451; page 26, line 521−525; page 28, line 558−564)**

---

## Author Response (AR2)

Dear Professor Xavier Querol,

Thank you very much for handling our manuscript submitted to *Atmospheric Chemistry and Physics* (**MS No.:** egusphere-2023-479; **Title:** Summertime response of ozone and fine particulate matter to mixing layer meteorology over the North China Plain).

We deeply thank the reviewers for giving constructive comments and suggestions that are very helpful to improve our manuscript. We have polished this paper by native English speakers, and carefully revised the manuscript with details showed below. We hope the revised manuscript meet the publication standards. To proceed, we have uploaded three files, including 1) our point-to-point reply; 2) the revised manuscript with changes highlighted in yellow; 3) the revised manuscript without track-changes.

On behalf of all the co-authors, I would like to thank you and referees for all the invaluable comments. Please feel free to contact me if you need any further information.

Yours Sincerely,

Jian Gao

Professor

Chinese Research Academy of Environmental Sciences, Beijing 100012, China

E-mail address: gaojian@craes.org.cn

Phone: 18911819868

**Anonymous referee #1:**
**Comment 1:** The text and the explanation of results have significantly improved, and the results are very relevant and interesting. However, the English used in the latest revision is much neglected compared to the original text, which sometimes hinders its understanding. It is strongly recommended to review the English.

**Answer:** Thank you for your approval and your valuable comments. According to your suggestions, we have polished this paper by native English speakers, and carefully revised the manuscript with details showed below. We hope the revised manuscript meet the publication standards.

**Comment 2:** Line 23: "opposite change characteristics"
Line 552: "opposite changes" is not understood.

**Answer**: "Opposite change characteristics" means the different effects of temperature on the change characteristics of $NO_3^-$ and $SO_4^{2-}$. $SO_4^{2-}$ concentration generally climbed up when temperature increased. Unlike $SO_4^{2-}$, which predominantly exists in the particle phase, $NO_3^-$ could be either presented as nitric acid ($HNO_3$) in the gas phase or as ammonium nitrate ($NH_4NO_3$) in the particle phase. Higher temperature could prompt the partitioning of nitrate to $HNO_3$, resulting in a significant decrease in $NO_3^-$ concentrations. To make this easier to understand, "opposite" has been replaced by "different" in our revised manuscript. (Page 1, line 24; Page 27, line 544)

**Comment 3:** Line 27: "superposition-composite effects"

**Answer**: The daily changes in MLH can form the interactive superposition influence effects on $O_3$ and $PM_{2.5}$. The superposition-composite effects has been reported before by Wang et al. (2022). It contains two meanings: the chemical interactions between $O_3$ and $PM_{2.5}$, as well as the accumulation of composite pollutants ($O_3$ and $PM_{2.5}$) along with the evolution of MLH. The formation of $PM_{2.5}$ and $O_3$ superposition-composite pollution along with the evolution of MLH was due to two superposition mechanism. First, the decrease in $PM_{2.5}$ during the daytime with the rise in MLH, can be partly offset by an increment in secondary pollutant formation derived from $O_3$ growth. Then, with the decrease in MLH at night, the concentration of the original existing $PM_{2.5}$ increased owing to unfavourable diffusion.

**Comment 4:** Line 83: "One speculative..." very long sentence

**Answer**: "One speculative reason for this phenomenon is that $PM_{2.5}$ does not reduce actinic flux and $HO_2$ radical significantly when the $PM_{2.5}$ concentration was low. " → "One possible reason is that when the $PM_{2.5}$ concentration is low, $PM_{2.5}$ does not reduce actinic flux and $HO_2$ radical significantly." (Page 4, line 83−84)

**Comment 5:** Line 109: "Observation was made..." -> "Observations were made"

**Answer**: This has been corrected in the manuscript. (Page 5, line 108)

**Comment 6:** Line 138: "1 hour", "1 h" or "hourly"

**Answer**: "1-h"→"1 hour". (Page 6, line 136)

**Comment 7:** The first paragraph of section 2.2 is confusing and not understood. It is proposed to summarize the reason for choosing MLH without getting into terminological discussions about PBL. The sentence in line 156 can be reformulated as: "The way the boundary layer describes the influences of air pollution is easily duplicated and confused.

**Answer**: Thanks to your valuable comments. We have deleted the terminological discussions about PBL, and the content in the first paragraph of section 2.2 has been reorganized in our manuscript. (Page 7, line 150–153).

**Comment 8:** In line 188, it is mentioned that the concept of MLH is clarified, but it is not the purpose of this work.

**Answer**: The purpose of this work was to investigate the impact of MLH on the change characteristics of ozone and fine particulate matter. We have rephrased this sentence in our revised manuscript. (Page 8, line 183–185)

**Comment 9:** Line 194, Typo in MDA8.

**Answer**: "MAD8"→"MDA8". (Page 9, line 189)

**Comment 10:** Line 200: "demonstrate?" Do the authors mean "show"?

**Answer**: "demonstrated"→"shown". (Page 9, line 195)

**Comment 11:** At the end of line 206, it would be advisable to add a period or separate the sentence.

**Answer:** "The $PM_{2.5}$ concentration was much lower comparing with ozone, with the mean, maximum, and minimum of the regional daily mean $PM_{2.5}$ concentration as 25.62 μg m$^{-3}$, 45.62 μg m$^{-3}$, and 11.32 μg m$^{-3}$, respectively, during the observation period."→"The $PM_{2.5}$ concentration was much lower comparing with $O_3$ during the observation period. The mean, maximum, and minimum of the regional daily mean $PM_{2.5}$ concentration was 25.62, 45.62, and 11.32 μg m$^{-3}$, respectively." (Page 9, line 199–202)

**Comment 12:** In expressions like "when MLH higher," a verb is missing. "When MLH was higher than..." This example is repeated in the manuscript.

**Answer**: We are sorry to make the mistakes, and the missing verbs have been added in our revised manuscript.

**Comment 13:** Line 222, It is obvious that comparing MDA8h with daily averaged $PM_{2.5}$ is not comparable.

**Answer**: According to your valuable suggestions, this sentence "we found that there is a lag time between the concentration peak of MDA8 $O_3$ and that of $PM_{2.5}$ along with the reduction of MLH" has been deleted in the revised manuscript.

**Comment 14:** Line 223, "turned to decline."

**Answer**: "turned to decline"→"declined". (Page 10, line 217)

**Comment 15:** Line 246, I recommend replacing "elsewhere" with "in other studies."

**Answer**: "elsewhere"→"in other studies". (Page 12, line 240)

**Comment 16:** Line 254, "higher" or "medium" boundary layer? The comparison with previous data is not very clear without numerical values.

**Answer**: The observations conducted by Reddy et al. (2012) showed that days of higher $O_3$ concentrations were associated with higher boundary layer height (2500 m) comparing with lower boundary layer height condition (1500 m) in April 1999, in India. Owing to the different observation seasons, the boundary layer height cannot be directly compared. This part has been deleted in our revised manuscript. (Page 12, line 244–247)

**Comment 17:** Line 261, "or not along" should be "along with."

**Answer**: "or not along"→"along with". (Page 12, line 253–254)

**Comment 18:** Line 273, Writing degrees like (45º-225º) appears as a range; it's better to remove the numerical value.

**Answer**: According to your comments, we have revised the expression of the degrees in the revised manuscript: "45–225°"→"45°–225°", "45–90, 90–135, 135–180, and 180–225°" → "45°–90°, 90°–135°, 135°–180°, and 180°–225°", "180–225°" → "180°–225°". (Page 12–13, line 266–273)

**Comment 19:** Subjective terms like "two obvious peaks" and "obviously" should be

avoided.

**Answer**: Thanks to your valuable comments. Subjective terms like "obvious" and "obviously" have bee deleted or replaced by "prominent" in our revised manuscript. (Page 17, line 354, 357; page 20, line 428; page 25, line 498; page 27, line 543)

**Comment 20:** Line 395, MLH, not BLH.

**Answer**: "BLH"→"MLH". (Page 19, line 388)

**Comment 21:** Line 424, I'm not sure what the authors want to convey with the values of SOR and NOR mentioned there

**Answer**: The oxidation ratio for sulfate (SOR, the molar ratio of sulfate to the sum of sulfate and $SO_2$) and nitrate (NOR, the molar ratio of nitrate to the sum of nitrate and $NO_2$) means the secondary conversion capacity of gaseous precursors to secondary aerosols. The high levels of SOR and NOR on $O_3$–$PM_{2.5}$CPD indicates the strong secondary formation of $SO_4^{2-}$ and $NO_3^-$ by high oxidation capacity, leading to the combined increase of $O_3$ and $PM_{2.5}$.

**Comment 22:** Line 560, A verb is missing.

**Answer**: The verb "of" has bee added in the revised manuscript. (Page 27, line 552)

**Comment 23:** Line 561, "is" instead of "was."

**Answer**: "was"→"is". (Page 27, line 553)

**Comment 24:** Line 562, "are" instead of "is."

**Answer**: "is"→"are". (Page 27, line 554)

**Reference**

Reddy, K. K., Naja, M., Ojha, N., Mahesh, P., and Lal, S.: Influences of the boundary layer evolution on surface ozone variations at a tropical rural site in India, J. Earth Syst. Sci., 121, 911-922, 10.1007/s12040-012-0200-z, 2012.

Wang, J., Yang, Y., Jiang, X., Wang, D., Zhong, J., and Wang, Y.: Observational study of the $PM_{2.5}$ and $O_3$ superposition-composite pollution event during spring 2020 in Beijing associated with the water vapor conveyor belt in the northern hemisphere, Atmos. Environ., 272, 10.1016/j.atmosenv.2022.118966, 2022.